# Systematic analysis of RNA-binding proteins identifies targetable therapeutic vulnerabilities in osteosarcoma

Yang Zhou[1,2,15], Partho Sarothi Ray[1,2,15], Jianguo Zhu[3], Frank Stein [3],
Mandy Rettel [3], Thileepan Sekaran[3], Sudeep Sahadevan [3],
Joel I. Perez-Perri [3], Eva K. Roth[1,2], Ola Myklebost [4,5],
Leonardo A. Meza-Zepeda[6], Andreas von Deimling [7,8], Chuli Fu[2],
Annika N. Brosig [1,2,3], Kjetil Boye [9], Michaela Nathrath [10,11,12],
Claudia Blattmann[12], Burkhard Lehner[13], Matthias W. Hentze [1,3] ✉ &
Andreas E. Kulozik [1,2,14] ✉

Osteosarcoma is the most common primary malignant bone tumor with a strong tendency to metastasize, limiting the prognosis of affected patients. Genomic, epigenomic and transcriptomic analyses have demonstrated the exquisite molecular complexity of this tumor, but have not sufficiently defined the underlying mechanisms or identified promising therapeutic targets. To systematically explore RNA-protein interactions relevant to OS, we define the RNA interactomes together with the full proteome and the transcriptome of cells from five malignant bone tumors (four osteosarcomata and one malignant giant cell tumor of the bone) and from normal mesenchymal stem cells and osteoblasts. These analyses uncover both systematic changes of the RNA-binding activities of defined RNA-binding proteins common to all osteosarcomata and individual alterations that are observed in only a subset of tumors. Functional analyses reveal a particular vulnerability of these tumors to translation inhibition and a positive feedback loop involving the RBP IGF2BP3 and the transcription factor Myc which affects cellular translation and OS cell viability. Our results thus provide insight into potentially clinically relevant RNA-binding protein-dependent mechanisms of osteosarcoma.

Osteosarcoma (OS) is the most common primary malignant bone tumor that is prevalent in children, adolescents and young adults. The current standard therapy calls for surgical removal of the primary tumor and clinically evident metastases in combination with pre-operative and postoperative chemotherapy[1]. Since the introduction of systemic chemotherapy during the 1970s and 1980s, the therapeutic outcome has stagnated, with a 5-year survival rate of approximately 60% for localized disease and only 20% for recurrence and/or metastatic disease[1,2]. Metastases are detected in around 20% of OS patients at diagnosis, and >80% of patients are presumed to have subclinical or undetectable micrometastases[3]. Therefore, there is an urgent medical need to broaden the understanding of the molecular mechanisms governing the oncogenesis of OS and to foster the development of innovative systemic therapies.

OS is characterized by transformed mesenchymal cells committed to the osteoblastic lineage, with the capacity of osteoid production[4]. Extensive studies on the genomic landscape of OS revealed a high level of genomic complexity and instability reflected by a high number of structural variants, frequent chromothripsis (massive genomic rearrangements occurring simultaneously, 77-89% of the

patients)[5,6], and kataegis (clusters of localized hypermutation, 50-85% of the patients)[7,8].The somatic alterations that are most commonly identified in OS involve a small set of genes (such as *TP53, RB1, ATRX,* and *DLG2*) for which no therapeutic options exist at present[7]. Overall, the complex and heterogeneous genomic signatures with few recurrent pharmacologically tractable alterations severely hamper the development of broadly applicable targeted therapies.

Genomic abnormalities may characterize, but do not fully reflect the pathogenicity of cancers. Post-transcriptional gene regulation is increasingly being appreciated for its contributions to the molecular mechanisms of cancers. RNA-binding proteins (RBPs) govern post-transcriptional gene regulation by controlling every aspect of RNA metabolism. Not surprisingly, alterations of RBPs have been implicated in every hallmark of tumor development[9,10]. An increasing number of studies have proved the potential of RBPs as promising therapeutic targets[9,10], but their role in OS has not yet been studied. A recent bioinformatics-based meta-analysis, utilizing gene expression data from OS samples and clinical data from osteosarcoma patients, generated an RBP-related prognostic signature for OS consisting of seven hub RBPs[11]. A number of transcriptomic studies have identified prognostic genes in OS, including *Myc* and *STC2*[12,13], but mechanisms regulating their expression have not been determined.

In the current work, we comprehensively characterize RNA-RBP interactions in OS in an unbiased and systematic fashion by using enhanced RNA interactome capture (eRIC)[14]. We present a comparison of the RNA interactomes of primary, patient-derived OS cells with mesenchymal stem cells and normal osteoblasts, which are thought to represent the cells of origin in OS. This comparison reveals RBPs with substantial quantitative RNA binding differences that may contribute to OS tumorigenesis. These include RBPs involved in enhanced global translation activity, mitochondrial translation, selective stabilization/translation of oncogenic transcripts, stemness maintenance and resistance to stress. Notably, we also identify inter-tumor heterogeneity of OS RNA interactomes with more aggressive tumors showing translation addiction and exquisite vulnerability for translation inhibition, as well as induction of an IGF2BP3-Myc positive feedback loop. Our research highlights the potential of RNA interactome analyses to unravel uncharted aspects of the molecular features of OS, thus opening perspectives for the development of new therapeutic strategies.

## Results
### Comprehensive and specific capture of the RNA interactomes of OS and normal bone/mesenchymal cells
RNA Interactome Capture (RIC) enables the comprehensive identification of the poly(A) RNA-binding proteomes of living cells[15]. This technology facilitates the discovery of non-canonical RBPs with unexpected RNA-binding activity, many of which have previously defined functions unrelated to RNA biology. RIC has been refined (enhanced RNA Interactome Capture = eRIC) with improvements of specificity and sensitivity by employing locked nucleic acids (LNA) in the capture process, allowing for further increases in the stringency of capture and background removal[14]. Hypothesizing that RBPs may play a previously unknown and targetable role in the oncogenesis of OS, we applied eRIC to define the RNA interactomes of OS and their normal mesenchymal/osteogenic cells of origin (osteoblasts (OB) and bone-marrow-derived mesenchymal stem cells (MSC)). Primary OS cells derived from four patients were generated either directly from clinical biopsies or from the tumor tissues grown in an orthotopic xenotransplanted mouse model that we have previously demonstrated to closely recapitulate the human disease[16]. These patient-derived cells were subjected to copy number profiling which confirmed features of OS such as highly complex copy number alterations and characteristic recurrent aberrations (e.g. amplification of *Myc* and loss of *RB1*) (Supplementary Fig. 1). These OS samples were complemented by primary patient-derived cells obtained from a malignant giant-cell tumor of bone (GCTB). This particular GCTB underwent malignant transformation with very aggressive clinical behavior similar to OS. GCTB is characterized by activation of the intercellular RANK/RANKL pathway in the tumor microenvironment and shows histological and pathogenic resemblance to OS with neoplastic components of mononuclear osteoblast-like stromal cells originating from MSCs[17,18].

For eRIC, cells were subjected to UV crosslinking (UV, 254 nm) or left untreated (noUV) as negative controls, followed by cell lysis and poly(A) RNA-protein complex capture, stringent washing and protein identification by mass spectrometry (Fig. 1a). Aliquots from the inputs and eRIC eluates were used for quality controls by protein analysis using SDS-PAGE followed by silver-staining or western blot, and for RNA analysis using a bioanalyzer. As expected, silver staining revealed that eRIC samples ( + UV) display profoundly distinct protein patterns compared to the inputs and the noUV controls, demonstrating a high degree of specificity and low background of the capture process (Fig. 1b). The western blots confirmed the successful capture of known poly(A)-RNA-binding proteins such as CSDE1, HuR and hnRNPK in the UV eluates. By contrast, abundant non-RNA-binding proteins such as α-tubulin and histone H3 included as negative controls were not captured by eRIC (Fig. 1c). The bioanalyzer profile revealed a typical length distribution of mRNAs and substantial depletion of rRNA in the population of captured RNAs (Fig. 1d). These results confirm the technical quality of the analyzed samples.

To define the RNA interactomes of patient-derived and normal cells, eRIC eluates (both UV and noUV conditions) from each sample were multiplexed using 16-plex tandem mass tag (TMT) and subjected to liquid chromatography–tandem mass spectrometry (LC-MS/MS). The corresponding input samples (full proteome, FP) were analyzed in a separate LC-MS/MS run. A total number of 6134 and 593 proteins were identified in the FPs and eRIC eluates, respectively (Supplementary Data 1). Proteins that were significantly enriched (at least 2-fold enrichment, $\log_2$ FC $\geq 1$ and an FDR < 0.05) in UV-crosslinked compared to no-UV controls were considered *high confidence* RBPs (Fig. 2a, Supplementary Data 1). We identified a total of 593 RBPs from all cells studied (OS, OB, MSCs), which we refer to as the bone/mesenchymal-cell RNA interactome, of which 583 RBPs were common to OB, MSC and OS cells (Fig. 2b). The RNA interactomes and the corresponding FPs correlate only weakly, indicating the specificity of RBP enrichment (Fig. 2c). Gene Ontology (GO) analysis revealed, as expected, that RNA-binding functions and RNA-biology related processes are highly overrepresented (Fig. 2d, Supplementary Data 2). Half of the bone/mesenchymal-cell RNA interactome proteins contain known canonical RNA-binding domains (RBDs) with RNA recognition motifs (RRMs) and KH domains as the most prevalent (Figs. 2e, f, Supplementary Data 2, 3). In line with previous RNA interactome studies[14,15,19], we also identified non-canonical RBPs lacking conventional RBDs, such as (metabolic) enzymes which account for about 15% (88 RBPs) of the RNA interactomes (Fig. 2e, Supplementary Data 3). In addition, intrinsically disordered protein regions (IDRs) of both known and unorthodox RBPs have been implicated in RNA binding[20]. In agreement with this observation, the analysis of IDRs showed a significantly higher disorder rank for the RNA interactomes relative to the same number of proteins randomly chosen from the FP (Fig. 2g). In comparison to previously published RNA interactomes, 33 RBPs were found exclusively in this bone/mesenchymal-cell RNA interactome and lack known RBDs (Supplementary Fig. 2, Supplementary Data 3). These proteins are enriched for regulators of bone homeostasis, metabolism, and related signaling pathways, such as collagen proteins (COL1A1, COL1A2, COL5A1, COL6A3), SMAD3 (a pivotal TGF-β signaling effector important for bone formation[21]), PLS3 (an important mediator for bone strength formation[22]), GIGYF1 (a modulator of insulin-like growth factor signaling pathway[23]), ANXA6 (a nucleator of bone mineralization[24]), STT3A (a glycosyltransferase for modification of glycoproteins which

constitute the most abundant non-collagenous proteins in bone matrix[25]). Several of these 33 RBPs have been linked to RNA-binding elsewhere, such as SMAD3[26] and GIGYF1[27], but their RNA-binding activity involved in bone cell biology has remained unknown. Overall, the bone/mesenchymal cell RNA interactome identified here largely resembles typical features of RNA interactomes from other origins and RBPs exclusively identified that likely associate with bone cell-specific characteristics. The RNA interactomes of OS and non-malignant bone and mesenchymal cells show substantial overlap. Therefore, we next assessed quantitative differences of RBPs between malignant bone tumors and their normal cells of origin.

## The RNA interactomes of osteosarcoma and normal bone/mesenchymal cells show systematic quantitative differences

To uncover RBPs linked to OS oncogenesis, we performed comparative analyses of the RNA interactomes of OS and their normal counterparts (OB/MSCs). RBPs with a fold change (FC) $\geq 1.5$ ($\log_2$ FC $\geq 0.6$) and an FDR $< 0.05$ were considered as "altered RBPs" (Supplementary Data 4). Notably, the number of altered RBPs is consistently higher when compared with OBs than with MSCs for all 4 OS samples (Fig. 3a), suggesting that these 4 osteosarcomata originate from cells at an early stage of bone development. The number of altered RBPs varies between the 4 osteosarcomata, with OSRH_2011/5 and I063_021 exhibiting more and OSKG and NRH_OS1 exhibiting fewer RBPs deviating from the normal cells (Fig. 3a). This variation correlates with the proliferation status of the OS cells (Fig. 3b) as well as the clinical aggressiveness (as defined by the presence of prognostically particularly unfavorable bone metastases) of the tumors. The more aggressive OSRH_2011/5 and I063_021, whose cells also proliferate faster in culture, showed more differentially binding RBPs than OSKG and NRH_OS1, respectively. The GCTB cells NRH_GCT1, which also show a high number of altered RBPs compared to OB/MSCs,

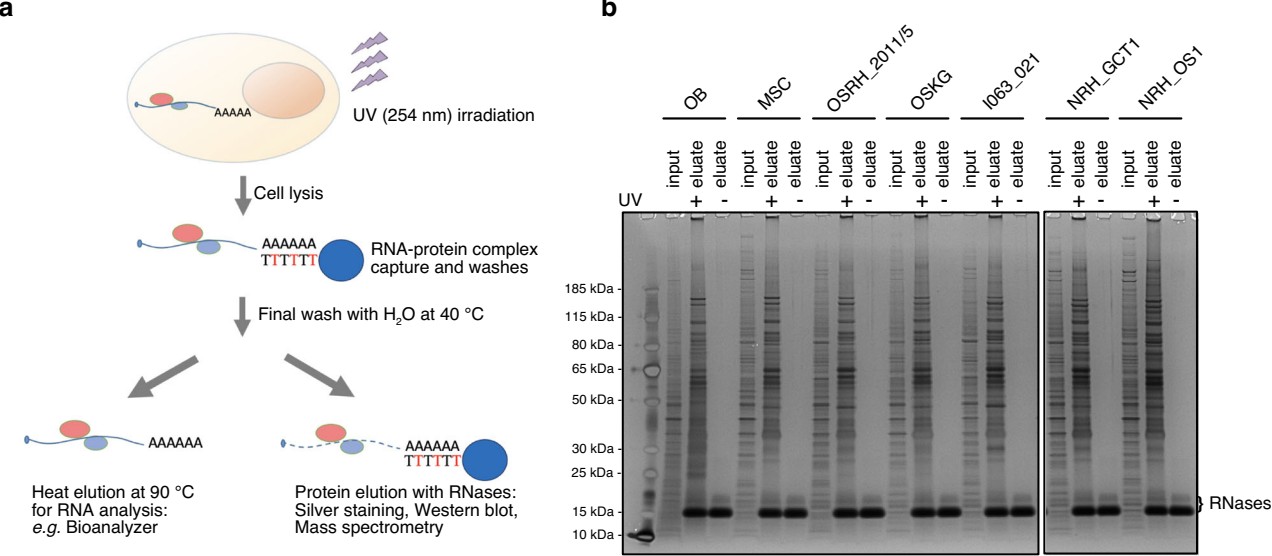

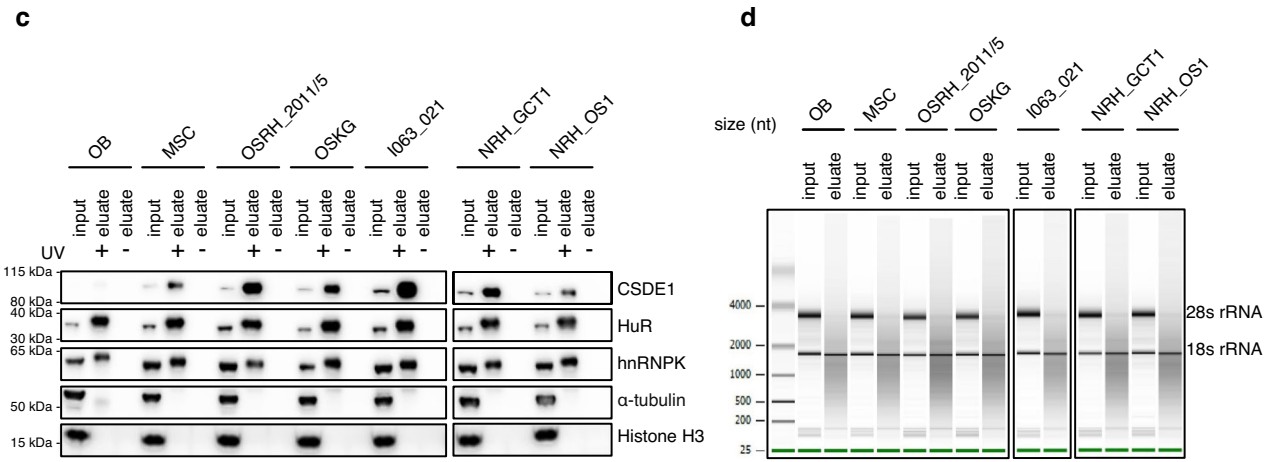

**Fig. 1 | RNA interactome capture of osteoblasts, MSC, and patient-derived sarcoma cells using eRIC. a** Schematic representation of RNA interactome capture by eRIC. "T" colored in red represents LNA thymidine. **b, c** RNA-protein complexes captured on LNA-coupled beads were eluted with RNases for protein analyses using silver staining (**b**) and western blot (**c**). Crosslinked samples are indicated as UV +, and non-crosslinked samples are indicated as UV -. Western blots include the known mRNA binding proteins CSDE1, HuR and hnRNPK, and the non-mRNA binding proteins α-tubulin and Histone H3 as negative controls. The silver staining and western blot were performed twice with the eRIC eluates and the FP aliquots from two biological replicates which showed similar results. **d** Input RNA and RNA captured by eRIC from non-crosslinked samples were analyzed by Bioanalyzer. Note that the RNA captured by eRIC shows the typical length distribution of mRNAs and substantial depletion of rRNA. The Bioanalyzer analyses were performed twice with the eRIC eluates and the FP aliquots from two biological replicates, which showed similar results. Abbreviations used in the figure: OB, osteoblasts; MSC, mesenchymal stem cells. Source data for blots are provided as a Source Data file.

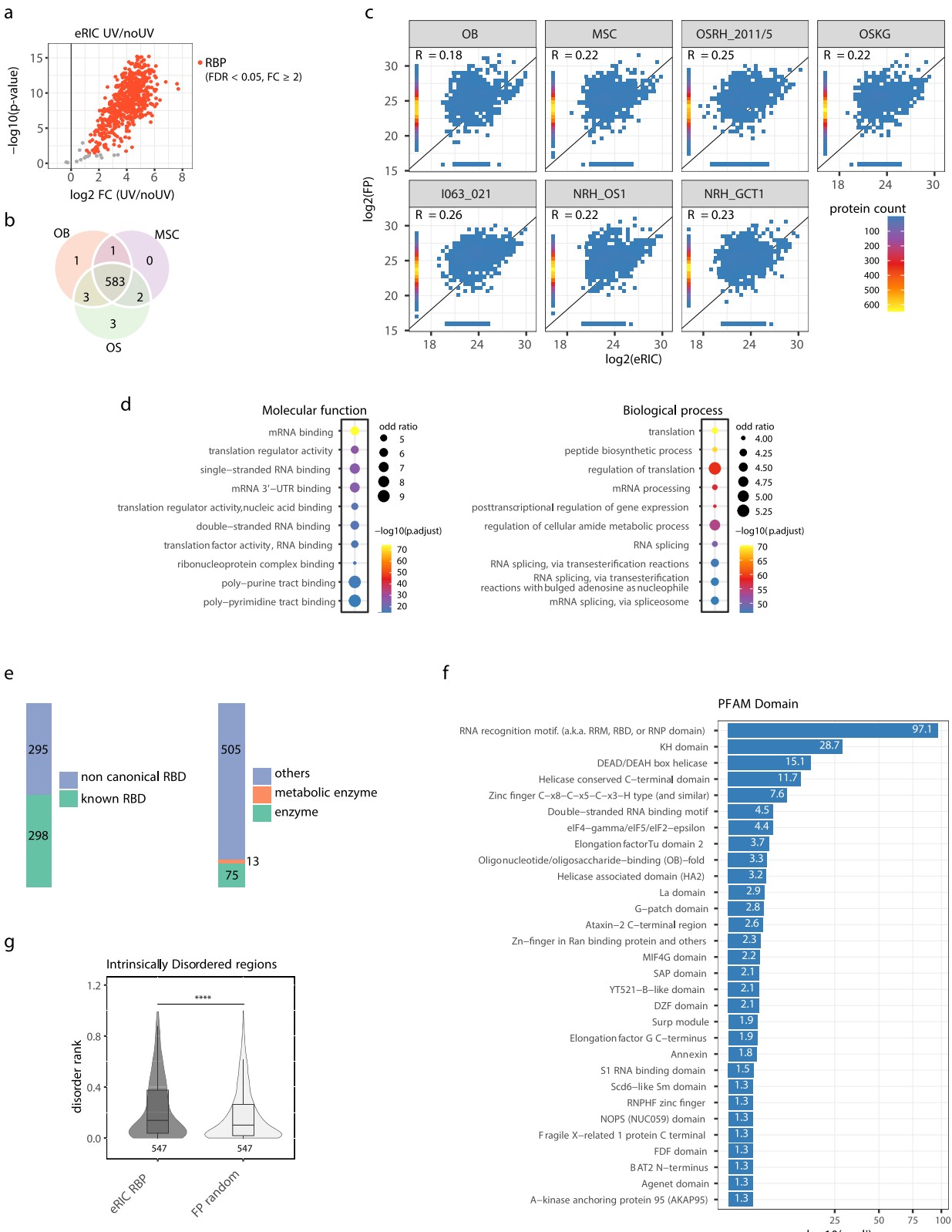

mirror the aggressive OS regarding their high proliferation rate and severe clinical behavior. Although the number of tumors analyzed here is small, these findings may nevertheless suggest that differences in the RNA interactome may reflect the phenotype of the more aggressive sarcomata.

We next compared the altered RBPs in each of the OS RNA interactome relative to the OB RNA interactome and found that several

RBPs were commonly changed in at least 2 OS samples, and of particular interest, 27 RBPs were systematically changed in all 4 OS with 6 of them being elevated and 21 being reduced (Fig. 3c, Supplementary Data 4). In comparison to osteoblasts, some of these systematically altered RBPs show the most striking differential binding (top-ranked 20%) in each OS RNA interactome (Fig. 3d). Three of the six elevated RBPs have been frequently reported for their oncogenic roles in

**Fig. 2 | Analysis of the RNA interactomes of osteosarcoma, normal mesenchymal stem cells and osteoblasts. a** Volcano plot depicting the $\log_2$ fold change (FC) of UV crosslinked (UV) over non-crosslinked (noUV) (x-axis) versus the *p* values (-log10; y-axis). The *p* values were obtained from the moderated t-statistic in the R package limma, after Benjamini-Hochberg adjustment. Proteins significantly enriched in the UV samples compared to the noUV controls with a FC ≥ 2 and an FDR < 0.05 were classified as high probability RBPs (red). Background non-RBPs are depicted in grey. **b** Venn diagram showing the overlap of RBPs identified in the OB, MSC and OS RNA interactomes. **c** Scatter plots showing the correlation between protein abundance (normalized TMT reporter ion intensities) in the UV crosslinked sample of the eRIC (y-axis, $\log_2$ transformed value) versus the corresponding full proteome (x-axis, $\log_2$ transformed value). These data show that the abundance of RBP in the interactomes does not correlate with protein abundance in the full proteomes thus demonstrating enrichment of RBPs by eRIC. **d** Gene ontology analyses for the identified 593 RBPs in the normal and malignant bone/mesenchymal-cell RNA interactomes with ten of the most significant overrepresented molecular function terms (left panel) and biological process terms (right panel). The *p* values were obtained from the one-sided version of Fisher's exact test in the R package Clusterprofiler, after Benjamini-Hochberg adjustment. **e** Number of the bone/mesenchymal-cell RNA-binding proteins that contain non-canonical RNA-binding domains (RBDs) and known RBDs, and that are known (metabolic) enzymes. **f** Analysis of protein domains using PFAM in DAVID (version 6.8) showing the significantly enriched (*p*.adj<0.05) domains in the RBPs identified in the bone/mesenchymal-cell RNA interactome. The *p* values were obtained from DAVID by using the one-sided version of Fisher's exact test, after Benjamini adjustment. **g** Box-and-whisker plot showing the disorder rank of the bone/mesenchymal-cell RNA interactome relative to an equal number of proteins randomly chosen from the full proteome (****$p$ = 8.4e-05). In the box-and-whisker plot, the line inside the box represents the median, while the lower and upper hinges of the box indicate the lower quartile (Q1) and upper quartile (Q3), corresponding to the 25th and 75th percentiles, respectively. The whiskers extend to a maximum of 1.5 times of Interquartile Range (IQR) beyond the box, and the lower and upper whisker ends represent the minima (Q1 − 1.5 * IQR) and maxima (Q3 + 1.5 * IQR), respectively. Outliers are not shown. This comparisonis based on the RNA interactomes and the full proteome data generated from 2 biological replicates. The *p*-value was obtained using the Wilcoxon test.

cancers. IGF2BP3 is upregulated in many tumor types and contributes to tumorigenesis and tumor progression by mediating mRNA stability and translation of malignancy-associated RNA targets[28]. YBX1, a multifunctional oncoprotein with both DNA- and RNA-binding capacity, has been reported to promote the proliferation and malignant phenotype of various cancer types and its overexpression often associates with poor prognosis of cancers, including OS[29]. MEX3A, a dual-function protein harboring both RNA-binding and E3 ligase activity, is also consistently upregulated in many cancers where it contributes to cancer development and progression, but little is known about the molecular mechanisms and the putative mRNA targets[30]. Importantly, there is limited published information about the function of RNA-binding in tumorigenesis for those RBPs which also exert non-RNA related functions such as DNA-binding and enzymatic activity. However, the data presented here indicate that alterations in their RNA-binding activity may contribute to the malignant phenotype.

Among the 21 RBPs showing systematic reductions of RNA binding in OS, several proteins have been shown to function as potential tumor suppressors in other types of cancers, including ANXA7[31], RBMX[32], RBM14[33], CSTB[34], and HDLBP[35]. Among these, HDLBP is of particular interest, because it encodes a multifunctional RNA-binding protein that has been linked to OS. HDLBP resides within a chromosomal region that is recurrently deleted in OS tumors. Consistent with a tumor suppressive function of this protein, its experimental depletion in the U2OS cell line increased colony formation[35]. Notably, the 21 RBPs exhibiting reduced RNA binding in OS are enriched for splicing factors or proteins involved in splicing, such as U2AF2, HNRNPR, HNRNPDL, RBM14, RBM15 and RBMX, suggesting that abnormal splicing plays a part in OS. Of note, some of the RBPs altered in OS (e.g., YBX1, IGF2BP3) also tend to differ in their RNA-binding behavior when comparing normal osteoblasts with MSCs, suggesting that these proteins may be related to the developmental stage and possibly to the stem cell function of MSCs. In fact, the role of YBX1 and IGF2BP3 in maintaining stemness has been reported[28,36,37], and IGF2BP3 has recently been shown to be enriched in the RNA interactome of human embryonic stem cells (hESCs) when compared to differentiated cells[38].

Despite the genetic heterogeneity of osteosarcoma, we thus identified systematic alterations of RNA-binding proteins in patient-derived osteosarcoma cells that may function as oncogenic or tumor-suppressive proteins. To further characterize these osteosarcomata using a multiomics approach, we performed whole transcriptome sequencing of the OS and GCT cells and identified differentially expressed RNAs in comparison to OB cells (Supplementary Fig. 3). The differentially expressed transcripts were compared with the differentially expressed genes at the proteome level (Supplementary Fig. 4) and significantly altered proteins identified in the RNA-interactome captures (Supplementary Fig. 5). This analysis identified genes that are differentially expressed either at the mRNA or the protein levels only (Supplementary Data 5, Supplementary Fig. 6, gold or blue), or in both (red), respectively and are also differentially enriched in the eRIC datasets (Supplementary Fig. 7). Most genes showed correlated changes at the transcriptomic and proteomic levels in the osteosarcomas and the giant cell tumor of bone. The top 20% of RBPs enriched in the RNA interactome of the OS and GCT cells, when mapped onto the transcriptomic and proteomic datasets showed that a majority of them exhibit enhanced abundance at both mRNA and protein levels. We also analyzed the genes which are upregulated in the proteome, but not in the transcriptome, indicating post-transcriptional control. OSRH_2011/5 and I063_21 show a substantially higher number of genes which are upregulated (FC ≥ 2, $\log_2$ FC ≥ 1) in the proteome (377 and 307, respectively) in comparison to OSKG and NRH_OS1 (124 and 103, respectively), with NRH_GCT1 exhibiting an intermediate number (196) (Supplementary Data 6). Among the genes that are highly upregulated in the proteomes of OSRH_2011/5, I063_021 and NRH_GCT1, but not in the transcriptome, are genes such as *TP53, PDCD4* and *MBD3* which are reported to be post-transcriptionally regulated and function as oncogenes and tumor-suppressor genes[39-41].

## RBPs affecting mRNA splicing are downregulated in the subgroup of proteins exhibiting differential RNA-binding independent of total protein abundance

We next asked the question of whether the quantitative differences in RNA-binding of the group of RBPs detailed above may be determined by increased protein abundance or also by differences in their RNA-binding activities. We thus performed an integrated analysis of alterations of these proteins in the RNA interactome and the full proteome (Fig. 4, Supplementary Data 7). The comparison of the RNA interactomes between OS and OB can be classified into 3 major groups: Group I contains RBPs showing unchanged RNA binding independently of changes of protein abundance (gray), Group II RBPs exhibit differential RNA binding in parallel to differential protein abundance (red), and of particular interest, Group III RBPs exhibit differential RNA binding without matched alterations in protein abundance (blue). While the RBPs that show the highest change in differential RNA binding also show the highest change in protein abundance (Group II), a subset of RBPs, which regulate mRNA splicing, including U2AF2, FUS, hnRNPs and RBM proteins, specifically exhibit reduced RNA binding without a change in abundance, suggesting that the RNA-binding function of these proteins specifically contributes to OS tumorigenesis. Although direct comparisons of alterations in protein enrichment in RNA-interactome capture with protein abundance in full proteome analysis is limited by technical constraints, a measure of "RNA-binding activity" of these proteins may be estimated by calculating the ratio

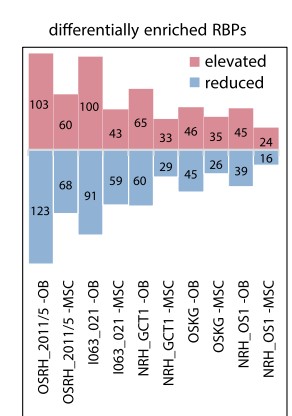

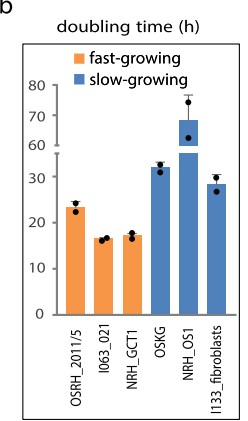

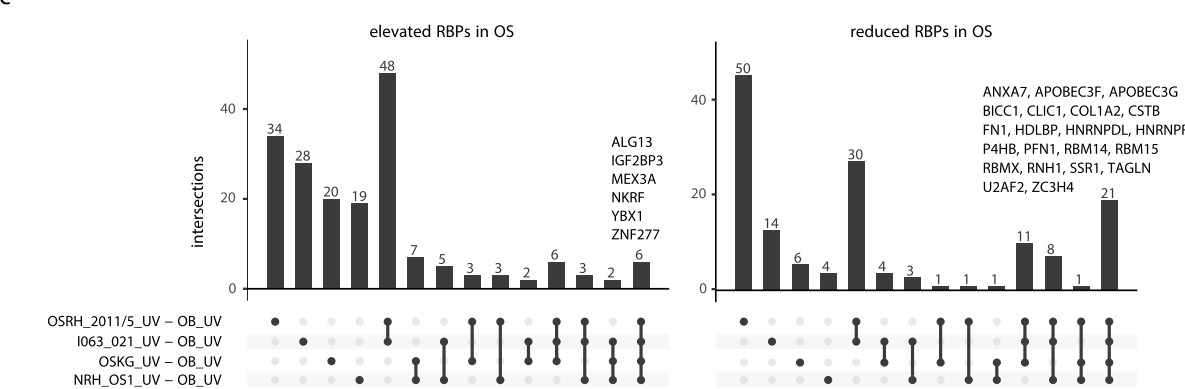

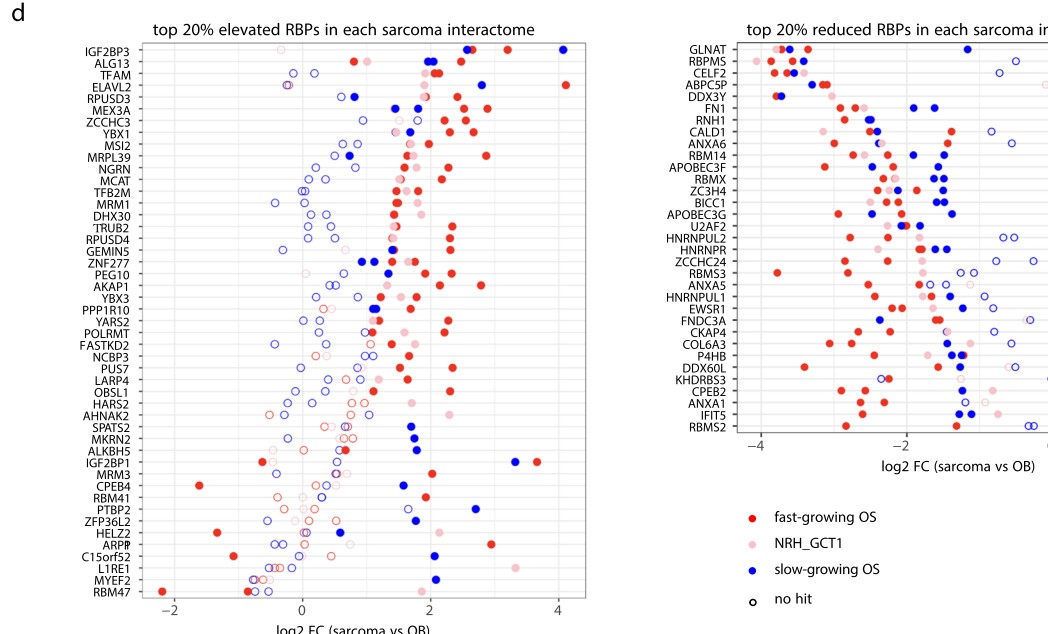

between the alteration of the protein capture in eRICs and the abundance in the full proteomes (Supplementary Data 7, last columns).

## Differences of the OS/GCTB RNA interactome correlate with clinical aggressiveness

Patients with osteosarcoma with bone metastases suffer from a particularly poor prognosis. Despite the generally aggressive nature of

osteosarcoma, we thus considered the tumors of patients OSRH_2011/5 and I063_021, who developed both lung and bone metastases in combination with mediastinal or liver metastases, respectively, to be particularly aggressive. Further, we also considered the rapidly progressive and malignant giant cell tumor of the bone in patient NRH_GCT1 to be particularly aggressive. Comparative analysis of the RNA-binding interactomes revealed RBPs with systematic, differential

**Fig. 3 | Comparative analyses of RNA interactomes reveal systematically altered RNA-binding proteins in bone tumors. a** Bar plot showing the number of significantly altered RBPs (FDR < 0.05 and FC ≥ 1.5 in the UV versus UV sample comparisons in the eRIC) in individual bone tumor RNA interactomes compared to OB/MSC. The number of elevated and reduced RBPs is indicated in pink and blue, respectively. **b** The malignant bone tumor cells can be grouped into a fast- and a slow-growing subgroup according to the population doubling time of the cells. Bars represent mean doubling time (h)± SD and dots represent individual data points for 2 biological replicates. **c** Upset plots showing the number of significantly

altered RBPs in each OS RNA interactome compared to the OB RNA interactome. The RBPs that are systematically altered in all 4 OS are shown with protein names. The terms "elevated" and "reduced" refer to the apparent RNA-binding activity from eRIC. **d** The most strongly (top-ranked 20%) altered RBPs in the RNA interactomes of each sarcoma compared to OB. The significantly altered RBPs (FDR < 0.05 and FC ≥ 1.5 in the UV versus UV sample comparisons) in each pairwise comparison were depicted in filled circles and unaltered RBPs were depicted in hollow circles. The elevated and reduced RBPs are shown in the left and right panel, respectively. Source data for graphs are provided as a Source Data file.

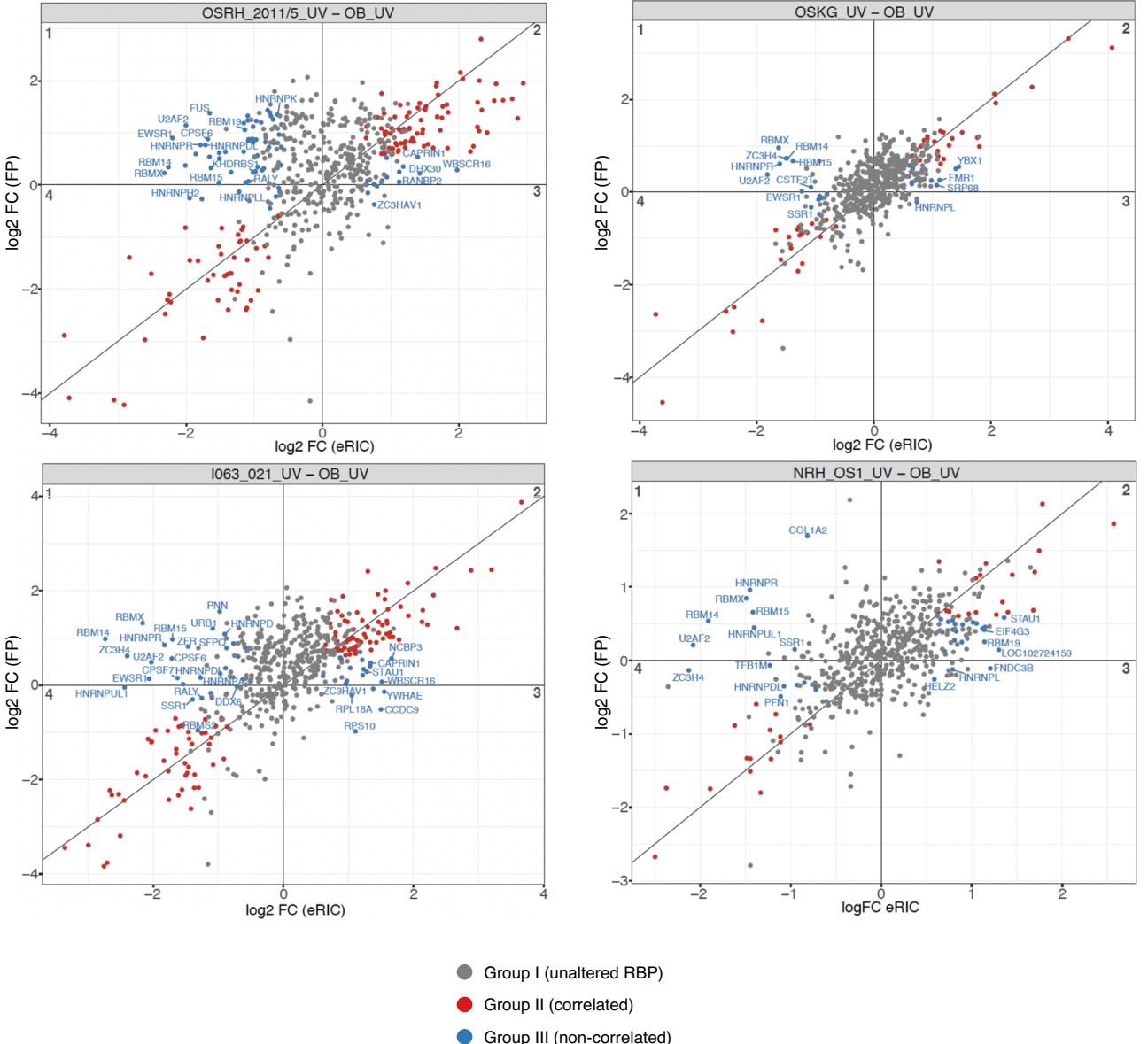

**Fig. 4 | Correlation between RNA-binding activity (as defined by protein abundance detected in eRIC) and total protein abundance in the full proteome (FP).** The relative RNA-binding activity reflected by UV sample comparisons (log₂ FC) in the eRIC (x-axis) is plotted against the relative RBP abundance (log₂ FC) in the full proteome (y-axis). Red dots indicate proteins whose changes of

RNA-binding correlates with changes of abundance in the FP in the indicated comparison. Blue dots indicate proteins whose RNA-binding is either decreased (quadrants 1 and 4) or increased (quadrants 2 and 3) in the indicated OS cells compared to OB. Gray dots indicate proteins showing unchanged RNA-binding independently of changes of protein abundance in the full proteome.

RNA-binding activity across the patient-derived OS cells analyzed here. In addition, other RBPs showed differential RNA-binding activity in cells derived from individual tumors. To globally characterize coordinately altered RBPs, hierarchical clustering was performed for differentially enriched RBPs in each pairwise comparison between OS and

OB/MSCs. For this comparison, the cells derived from the highly aggressive patient-derived malignant GCTB were also included. Based on the quantitative differential RBP enrichment in the OS/GCTB RNA interactome, 7 major clusters were defined. RBPs in clusters 1 and 2 were generally elevated, while the others were reduced in the

interactomes of OS cells (Fig. 5a). Notably, a clear distinction emerges between the highly aggressive tumors and the less aggressive ones: the clinically more aggressive OSRH_2011/5, I063_021 and NRH_GCT1 display RBP activity patterns that are distinct from the less aggressive OSKG and NRH_OS1. Accordingly, the three clinically particularly aggressive sarcomata cluster together, whereas the two less aggressive tumors display more similarities with normal osteoblasts and mesenchymal stem cells (Fig. 5b). The distinction between the clinically more aggressive, fast-growing OS and the less aggressive, slow-growing OS is also observed at the overall gene expression level, both the transcriptome and the proteome. Principal component analysis (PCA) plots of the OS and GCTB cells' transcriptomes, proteomes and RNA interactomes, together with that of OB and MSC, show that the more aggressive OSRH_2011/5 and I063_021 OS are more closely related, and distinct from that of the less aggressive OSKG and NRH_OS1, which also cluster closer to the MSC and OB samples (Fig. 5c). Interestingly, the GCTB sample NRH_GCT1 is distinct from the more aggressive OS cells at the level of gene expression, suggesting that the high growth rate and the clinical aggressiveness of this tumor are governed by other mechanisms than in the aggressive osteosarcomata. The PCA plots therefore indicate distinct evolutionary trajectories of divergence from the normal cells of origin in the more aggressive and less aggressive subtypes of the OS and the GCTB.

### Emerging roles of RBPs in osteosarcoma

To functionally characterize RBPs with differential RNA-binding activity in OS, we performed GO analysis on protein clusters derived by hierarchical clustering (Fig. 5d). The RBPs in cluster 1 and 2 (elevated in all OS cells) are enriched for mitochondrial translation and translation-related proteins, while reduced proteins (cluster 3-7) are enriched for RNA splicing, and mRNA processing functions (Fig. 5d, Supplementary Data 2). With a closer look at the elevated mitochondrial translation-related proteins, we observed that these RBPs are mitochondrial ribosomal proteins directly involved in translation, as well as RBPs involved in mitochondrial RNA modifications (TRUB2, RPUSD3, TRMT10C), processing (PNPT1), stability (TBRG4, SUPV3L1, LRPPRC) and regulation of translation (LRPPRC, GRSF1) (Fig. 6a). These proteins also include the cytosolic mRNA-binding protein CLUH, which interestingly binds mRNAs of nuclear-encoded mitochondrial proteins and regulates their localized translation close to mitochondria[42–44]. These proteins are predominantly elevated in the interactomes of the more aggressive sarcomata.

The abundance of several of these proteins is known to be increased in various other tumors relative to their corresponding normal tissues, including MRPL38, MRPS27, TFB1M, GRSF1[45], as well as LRPPRC, the overexpression of which has been associated with poor prognosis[46]. Consistent with the findings reported here, globally enhanced levels of mitochondrial proteins have also been seen in mesothelioma and contributing to abnormal mitochondrial morphology and metabolic changes providing energy and metabolites for tumor cell growth[47]. In comparison to OB, we also observed enrichment of mitochondrial translation-related proteins in the interactome and in the full proteome of MSCs (Fig. 6a). This is in line with a recent report showing a cluster of elevated mitochondrial translation-related proteins in the full proteomes of MSCs relative to OB[48]. These observations suggest differential mitochondrial protein function in association with stemness. In fact, mitochondria have been reported to play a pivotal role in maintaining stemness by providing energy and through the biosynthesis of key metabolites in both normal tissues and cancer stem cells[49,50].

Another large cluster of altered RBPs includes cytoplasmic translation-related proteins, of which ribosomal proteins, translation initiation and translation termination factors are generally elevated more strongly in the more aggressive sarcomata (Fig. 6b), indicating more active global translation in these cells. In addition to proteins involved in global translation, several translation-related RBPs with differential RNA-binding activity are known to regulate the stability and/or translation of selective mRNAs by binding to specific sequence/structure elements or RNA modifications primarily in 5' and 3' UTRs of mRNAs. This class of proteins is exemplified by PUM1 and PUM2, the La proteins (LARP1, LARP4, LARP4B), the IGF2BP family (IGF2BP1 and IGF2BP3), and the Y-box proteins YBX1 and YBX3 (Fig. 6b), most of which have been implicated in tumorigenesis and as hallmarks of cancer by targeting oncogenic/tumor suppressor transcripts[9,51]. Interestingly, IGF2BP3 is found to be the most enriched RBP in the RNA interactome, and is also abundant in the full proteome, of the osteosarcomata, but not in the GCTB. Although the data presented here are limited by the small number of patients, these data indicate that both global and oncogenic translation-related RBPs are enhanced in osteosarcoma, particularly in the more aggressive tumors developing multifocal bone, liver and mediastinal metastases.

To further explore whether the enrichment of RBPs related to translation and mRNA metabolism, especially in the RNA-interactomes of the more aggressive OS, is reflected in the gene expression programs of these cells, we performed gene ontology (GO) analysis of the significantly upregulated genes in the transcriptomes and proteomes of OS and GCTB samples compared to OB (Fig. 6 c and d). Remarkably, the most enriched GO terms in the more aggressive OS, OSRH_2011/5 and I063_021, are related to RNA metabolism and translation (e.g. ribonucleoprotein biogenesis, noncoding RNA processing, ribosome biogenesis, mRNA processing, rRNA processing and splicing), whereas such terms and pathways are notably absent in the less aggressive OS, OSKG and NRH_OS1. These results indicate that the enrichment of RBPs involved in RNA translation and metabolism in the more clinically aggressive OS may impart a more translation- and RNA metabolism-centric gene expression program to these osteosarcomata.

We also investigated whether the enrichment of specific RBPs in the RNA interactomes correlated with enhanced expression of their target mRNAs in the transcriptome, indicating RBP-mediated stabilization of these transcripts. For this we selected three RBPs which are among the top 20% most elevated in the sarcoma interactomes (Fig. 3d): IGF2BP3, MEX3A and AKAP1. Among these RBPs, IGF2BP3 is highly elevated in the interactomes of all the OS, but not the GCTB. MEX3A is elevated in all the OS and the GCTB, whereas AKAP1 is most highly elevated in the two aggressive OS, and then in the GCTB, but not in the less aggressive OS. Using three orthogonal datasets for the mRNA targets of these three RBPs (an intersection of CLIPseq targets and downregulated transcripts upon shRNA-mediated knockdown of IGF2BP3[52] (Supplementary Data 8), RIPseq targets of MEX3A[53] and eCLIPseq targets of AKAP1 from the ENCODE database (https://www.encodeproject.org/genes/8165/)), we interrogated the expression levels of their targets in the transcriptomes of the OS and GCTB cells. In case of IGF2BP3, we find the highest number of upregulated target transcripts in the two aggressive OS, I063_021 and OSRH_2011/5 and in OSKG (Supplementary Fig. 8), all three of which show the highest enrichment of IGF2BP3 in the RNA interactome (Fig. 6b, Supplementary Figs. 5, 9). Upregulated MEX3A targets are nearly similar in number in the OS and GCTB cells, except in the less aggressive OS, NRH_OS1. For AKAP1, the highest number of upregulated targets are in the aggressive OS I063_021 and OSRH_2011/5, followed by the GCTB, NRH_GCT1, and lowest in the less aggressive OS, OSKG and NRH_OS1, showing a correlation with the enrichment of AKAP1 in the RNA interactome of these cells (Fig. 3d). Together, the level of enrichment of the selected RBPs was found to be correlated with the upregulation of their target transcripts in the sarcomata.

### High translation rates of aggressive sarcomas represent a potential therapeutic vulnerability

The enrichment of translation-regulatory RBPs in the RNA interactome, and the corresponding upregulation of translation-related

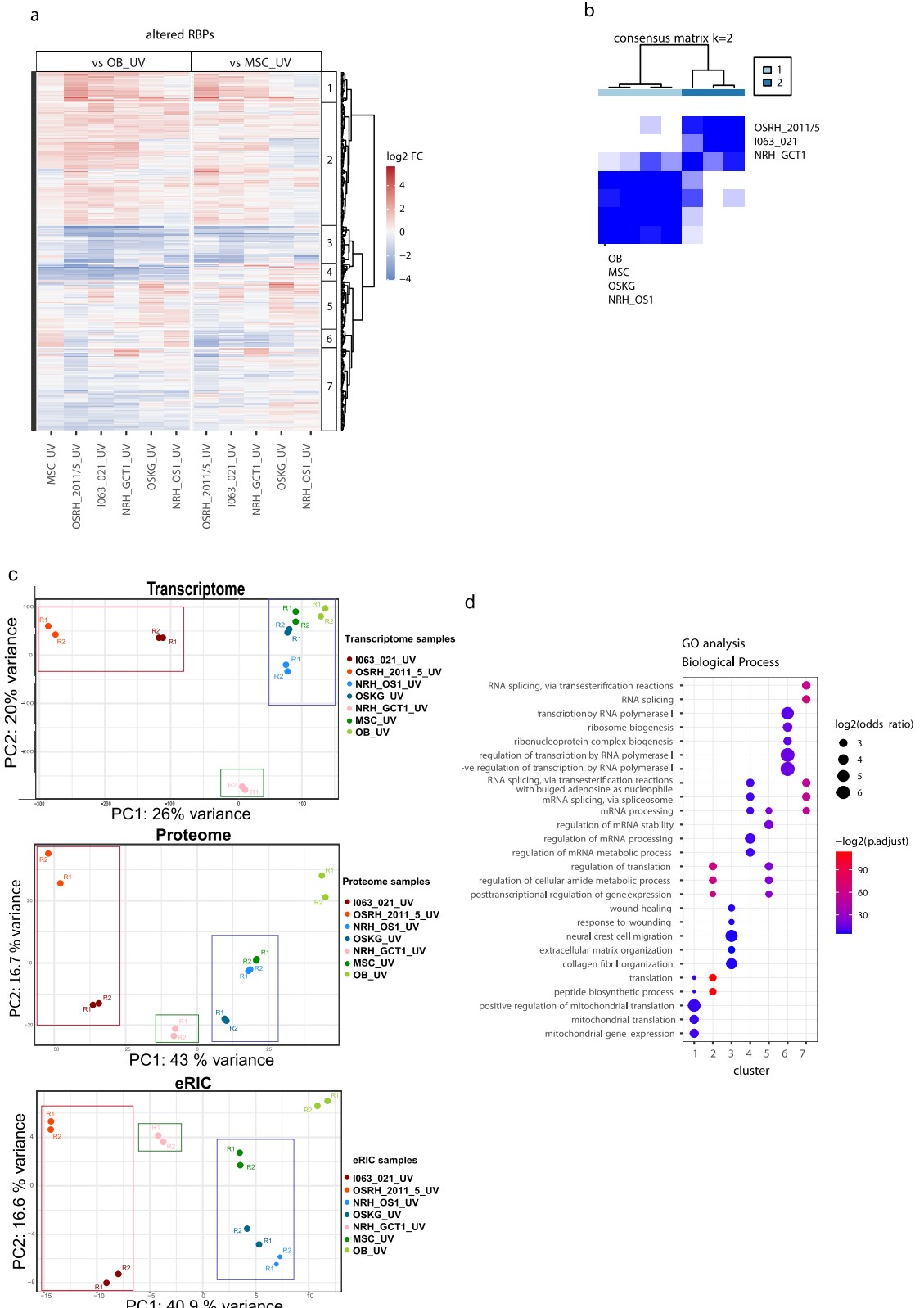

gene expression in the transcriptome and proteome, of the more aggressive sarcomata led us to hypothesize that these sarcomas are more active in protein synthesis. We directly validated this hypothesis by measuring global protein synthesis using metabolic labeling (Fig.7a). The two clinically aggressive osteosarcomata OSRH_2011/5 and I063_021, together with the aggressive GCTB, NRH_GCT1, exhibited higher levels of protein synthesis compared to the less aggressive

OSKG and NRH_OS1 and the non-neoplastic stromal cells derived from OS tumor tissue I133 (Fig. 7a). Interestingly, OSRH_2011/5 and I063_021 proved to be more vulnerable to inhibition of translation, showing strongly reduced cell viability when treated with sub-lethal doses of the translation inhibitor cycloheximide (CHX) (Fig. 7b). Notably, the effect of CHX on cell viability was more pronounced in these two clinically aggressive osteosarcomata than the less aggressive OSKG

**Fig. 5 | Comparative analyses of RNA interactomes reveal translation and RNA splicing/processing as differential functional categories in osteosarcoma.**
**a** Heat map showing the relative RBP abundance (log₂ FC among UV crosslinked samples in eRIC) in the comparative RNA interactome analysis in each indicated sarcoma in comparison with either OB or MSC (FDR < 0.05, FC ≥ 1.5), and in MSC compared to OB (FDR < 0.05, FC ≥ 1.5). Color indicates the log₂ FC. The resulting 350 differentially enriched proteins were clustered hierarchically resulting in 7 main clusters. **b** Consensus analysis based on the RBP abundance in RNA interactomes (UV samples from eRIC) shows the classification of sample subgroups. Subgroups of samples were identified by hierarchical consensus clustering using the "ConsensusClusterPlus" R package. The color gradients indicate consensus values from 0 (never clustered together, white color) to 1 (always clustered

together, dark blue). The color bar indicates cluster 1 (light blue) and cluster 2 (dark blue), respectively. The output report shows that a cluster number (k) of 2 is optimal. **c** Principal component analysis (PCA) plots of the transcriptome, proteome and RNA interactome (eRIC) of the OS, GCTB, OB and MSC. R1 and R2 represent the two replicates of each experiment for each cell type. The coloured boxes (red, blue and green) indicate the differential clustering of the fast-growing OS, the slow-growing OS and MSC and the GCTB cells respectively in the PCA plots. **d** The GO analysis of each of the 7 clusters defined in panel **a** shows the most significantly enriched (p.adj<0.05, and top 10) biological process terms. The p values were obtained from the one-sided version of Fisher's exact test in the R package Clusterprofiler, after Benjamini-Hochberg adjustment.

and NRH_OS1, the GCTB NRH_GCT1, the I133 fibroblasts and fast-growing HeLa cells (Fig.7b).

We further interrogated this vulnerability of the OS to translation inhibition by employing homoharringtonine (HHT), a translation inhibitor FDA-approved for the treatment of chronic myeloid leukemia (CML)[54]. HHT effectively inhibited protein synthesis in the OS cells, with a more pronounced effect on the OS with highly active protein synthesis, but not in the GCTB cells (Fig. 7c). HHT reduced cell viability in a dose-dependent manner and was highly active at concentrations observed in patients treated with HHT[55]. HHT was more effective in reducing the viability of the "translation-hungry" OS cells compared to cells characterized by less active protein synthesis and, remarkably, also of the translationally highly active GCTB cells (Fig. 7d). These data indicate that response to HHT is specific for the highly aggressive and translationally active osteosarcomata, whereas the equally aggressive and translationally active malignant GCTB did not exhibit a similar response to HHT. Therefore, the response to HHT does not represent a mere non-specific effect on translationally active tumor cells.

## An IGF2BP3/Myc positive feedback loop constitutes an oncogenic signature of OS with highly active translation

To further explore the oncogenic signature of RBPs enriched in the OS-RNA interactomes, the "hallmark" enrichment analysis was performed based on the "hallmark" gene sets sourced from the Molecular Signatures Database (MSigDB). From the hallmark gene sets defined in this database, 3 are significantly enriched in the RNA interactomes of more aggressive but not the less aggressive osteosarcomata, including 2 sets of RBPs that are Myc-targets (Fig. 8a). This observation is consistent with the amplification of the *Myc* gene in the more aggressive OS with high translation rates, OSRH_2011/5 and I063_021 (Supplementary Fig. 1). Transcriptional regulation of RBPs by Myc in shaping the RNA interactome has also been reported in mouse embryonic stem cells (mESCs). The mESC RNA interactome intersects significantly with the Myc module but not with other modules of the ESC transcription program[56]. The Myc module is pervasively active in cancers as well, and represents the most shared similarity of ES- and cancer cell signatures[57]. Consistent with the particular role of Myc in regulating the expression of RBPs involved in oncogenesis, the comparison with the RNA interactome of normal osteoblasts reveals that RBPs that are Myc-targets are enriched in the RNA interactomes and proteomes of the more aggressive osteosarcomata (Fig. 8b).

Conversely, *Myc* mRNA is unstable[58], and several RBPs identified here to be enriched in the OS interactomes have been previously reported to enhance the stability and/or translation of *Myc* mRNA, including YBX1[59,60], IGF2BP3[52,60,61], CSDE1[62], CAPRIN1[63], PABPC1[64] and FXR1[65]. In our subsequent analyses, we focused on IGF2BP3, the most highly enriched RBP in the RNA interactomes of the OS, which is also a known regulator of *Myc* mRNA stability and translation[52]. Conversely, Myc has been reported to be a transcriptional activator of IGF2BP3[66].We hypothesized a positive feedback loop connecting IGF2BP3 and *Myc*, wherein IGF2BP3 enhances *Myc* expression post-transcriptionally, while Myc transcriptionally enhances IGF2BP3

expression. IGF2BP3 was found to be highly expressed in the two OS with highly active translation, OSRH_2011/5 and I063_021, and also in OSKG, a less aggressive OS (Fig. 8c, upper panel). Myc protein expression was high in OSRH_2011/5 and I063_021 but not in OSKG (Fig. 8c, middle panel), consistent with the *Myc* gene amplification in the former (Supplementary Fig. 1). Notably, IGF2BP3 expression was undetectable in the aggressive GCTB, NRH_GCT1, mirroring the distinct molecular mechanisms governing aggressiveness in this bone tumor. We have next tested the prediction of the hypothesized positive feedback loop that IGF2BP3 increases Myc expression, by depleting IGF2BP3 by siRNA transfection in OSRH_2011/5, I063_21 and OSKG cells (Fig. 8d). Depletion of IGF2BP3 resulted in substantially decreased Myc protein expression in the aggressive OS, OSRH_2011/5 and I063_21, demonstrating the post-transcriptional regulation of *Myc* by IGF2BP3 in these cells, even in the background of *c-Myc* gene amplification. The interaction of IGF2BP3 with *c-Myc* mRNA was validated by RNA-immunoprecipitation using an IGF2BP3 antibody. This analysis revealed the interaction of *c-Myc* mRNA with IGF2BP3 protein in OSRH_2011/5 and I063_21 cells, while it was nearly undetectable in OSKG cells (Fig. 8e, left panel). Depletion of IGF2BP3 also resulted in decreased *c-Myc* mRNA in these cells, but not in OSKG, supporting the role of IGF2BP3 in *c-Myc* mRNA binding and stabilization (Fig. 8e, right panel). IGF2BP3 depletion also resulted in stronger reduction of translation in the aggressive OS I063_021 compared to OSKG, and also sensitized it further to the translation inhibitors HHT and CHX (Fig. 8f).Taken together, these data validate the predicted positive post-transcriptional regulation of Myc by IGF2BP3, the most strongly enriched RBP in the RNA interactome of the aggressive osteosarcomata.

We next tested the second arm of the proposed positive feedback loop by treating OS cells with volasertib, a clinically approved inhibitor of Polo-like kinase 1 (Plk1). Plk1 specifically binds to and phosphorylates the SCF^Fbw7 ubiquitin ligase, and promotes its autopolyubiquitination and proteasomal degradation, countering Fbw7-mediated degradation of Myc[67]. Volasertib-mediated inhibition of Plk1 therefore destabilizes Myc by blocking the phosphorylation of Fbw7 by Plk1 leading to its stabilization, thus reducing the abundance of the Myc protein. Treatment of the OS cells with volasertib resulted in the expected decrease of Myc protein expression (Fig. 8g, upper panel). Consistent with the transcriptional regulation of IGF2BP3 by Myc, this treatment also reduced IGF2BP3 expression both at the protein and RNA level in a dose-dependent manner (Fig. 8g lower panel, Fig. 8h). Volasertib treatment also reduced the viability of OS cells in a dose-dependent manner, with minimal effect on NRH_GCT1, which lack the IGF2BP3-Myc positive feedback loop, or I133 fibroblasts and HeLa cells (Fig. 8i). Taken together, these data indicate that the positive feedback loop involving the RBP IGF2BP3 and the transcription factor Myc represent a vulnerable oncogenic signature of osteosarcoma (Fig. 8j).

## Discussion
RNA interactome analysis has enriched the repertoire of comprehensive omics technologies suitable to unravel tumor biology. The

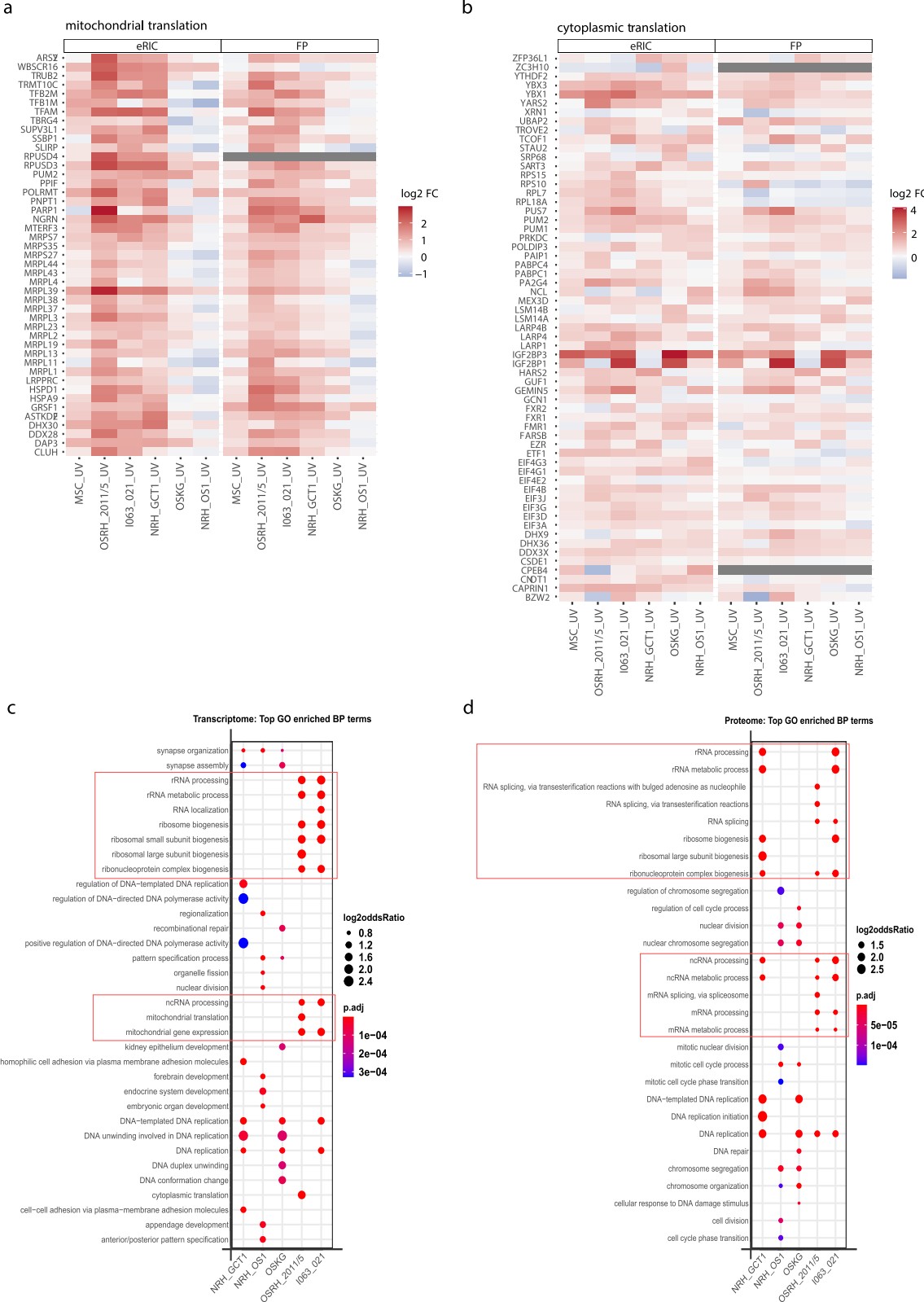

**Fig. 6 | Enrichment of RBPs involved in mitochondrial and cytoplasmic translation in the fast-growing OS imparts translation-centric gene expression programs. a, b** Heatmap showing the relative RBP abundance (log₂ FC) of 45 mitochondria-related proteins (**a**) and 61 cytoplasmic translation-related proteins (**b**) in comparison of sarcomata with OB in the RNA interactome (eRIC) and in the full proteome (FP). **c, d** GO analysis of significantly upregulated genes (log₂ FC ≥ 1) in the transcriptomes and proteomes of the sarcomata in comparison to OB shows the most significantly enriched (p.adj<0.05, and top 10%) biological process (BP) terms. The *p* values were obtained from the one-sided version of Fisher's exact test in the R package Clusterprofiler, after Benjamini-Hochberg adjustment. The red boxes in **c** and **d** denote BP terms related to RNA metabolism and translation, specifically enriched in the fast-growing OS and GCTB transcriptomes and proteomes respectively.

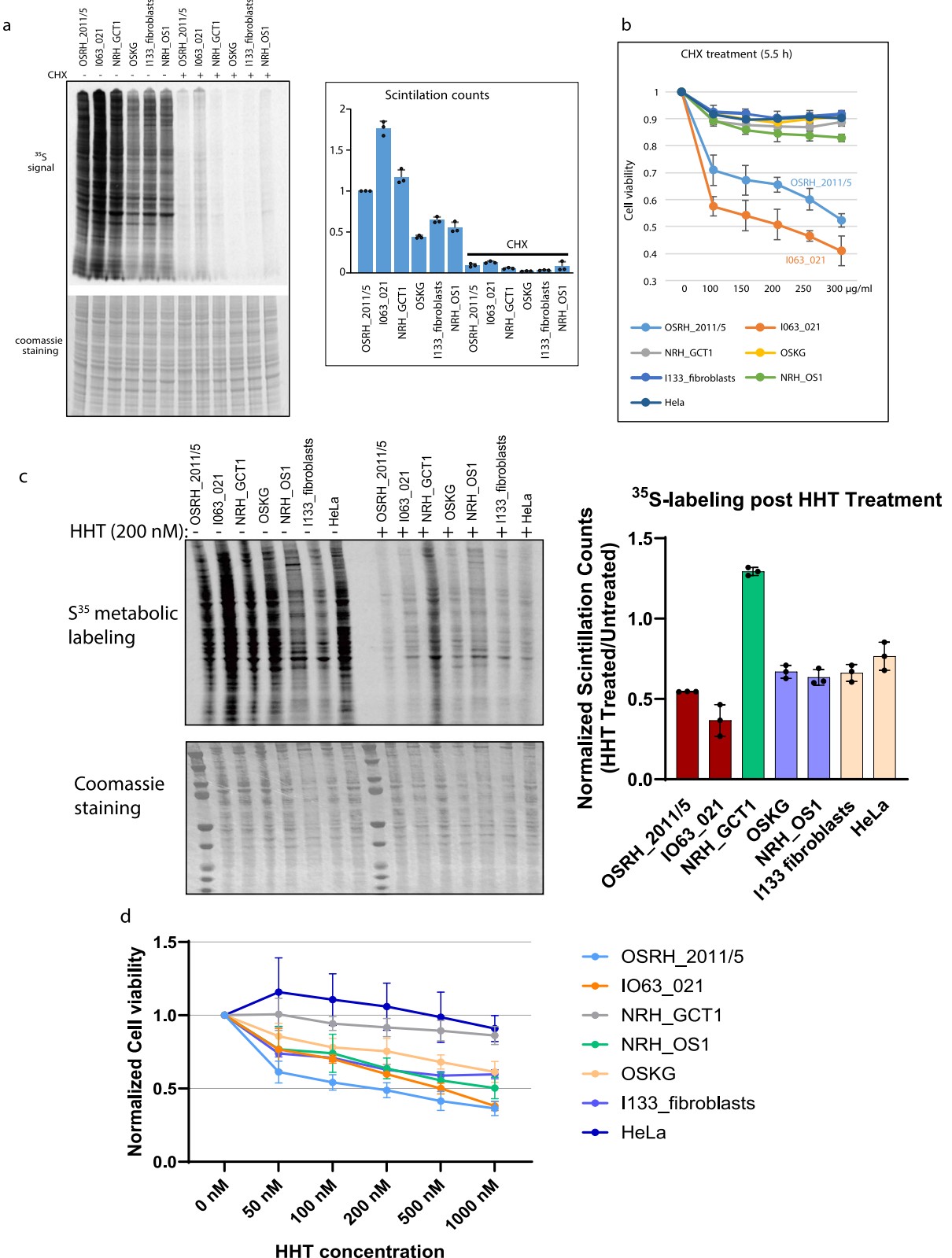

previously employed methods have been genome- or proteome-centric, which left a gap in the assessment of the many posttranscriptional steps in biology, the critical connecting points between genomes and proteomes. In this report, we fill this gap by the comprehensive analyses of the RNA interactomes of patient-derived osteosarcoma cells, compared to those of non-malignant osteoblasts and mesenchymal stem cells. Although the OS and the OB/MSCs RNA interactomes share

most RBPs, they reveal substantial quantitative differences, which characterize the tumor-specific RNA interactomes of osteosarcoma. However, many tumors exhibit intricate subclonal architectures, and it remains possible that the tumor cells analyzed in this study originate from a particular subclone of the primary tumor which might influence the comparison between the OS and the OB/MSCs and between the different osteosarcomata.

**Fig. 7 | Fast-growing OS have high translation activity and are more vulnerable to translation inhibition. a** Measurement of global protein synthesis in absence and presence of cycloheximide treatment by metabolic labeling with [35]S-methionine/cysteine. Autoradiography of [35]S signal (upper left panel) and coomassie staining of the gel (lower left panel). [35]S-methionine/cysteine incorporation in proteins was quantified using scintillation counting (right panel). The scintillation counts of cell lysates were determined and normalized to the total protein amount. The scintillation counts of OSRH_2011/5 were set to 1. Cells were treated with cycloheximide (CHX, 50 μg/ml) for 2 h prior the assay. Data represent mean ± standard deviation (SD) derived from three biological replicates. **b** Cell viability, as measured by cell titer blue assay for sarcoma cells, non-neoplastic stromal cells (I133_fibroblasts) derived from an OS tumor tissue and the HeLa cell line after treatment with the translation inhibitor CHX up to 300 μg/ml for 5.5 h. Data represent mean ± SD derived from three biological replicates. **c** [35]S metabolic labeling of cells in absence and presence of HHT treatment. The [35]S metabolic labeling was done as in **a**. Cells were treated with homoharringtonine (HHT, 200 nM) for 2 h prior to [35]S labeling. [35]S incorporation in proteins was quantified using scintillation counting (right panel). Scintillation counts from HHT-treated cells are normalized to scintillation counts from untreated cells for each cell type. Data represent mean ± standard deviation (SD) derived from three biological replicates. **d** Cell viability, as measured by Celltitre-Glo assay, after treatment with HHT (0–1000 nM) for 24 h, normalized to untreated cells. Data represent mean ± SD derived from three biological replicates. Source data for graphs and blots are provided as a Source Data files.

Our comparative analyses reveal that the OS RNA interactomes exhibit more similarity to MSCs than to osteoblasts (Figs. 3a, 5a, b). Several strongly enriched RBPs in the OS-OB but not in the OS-MSC comparison have been implicated in sustaining stemness in cancer cells. This group of proteins include YBX1[36], IGF2BP3[28,37] and MEX3A[30], which are elevated in all 4 OS, and IGF2BP1[37], MSI2[68] and Pum2[69] which are enriched in at least 2 OS (Figs. 3c, d, 6b). Among these, IGF2BP1/3 and MEX3A were reported to be expressed in an oncofetal fashion. Therefore, these data indicate that the osteosarcomata studied here originate from an early stage of lineage commitment sharing features with normal mesenchymal stem cells.

We identify shared oncogenic signatures of all 4 OS RNA interactomes. These notably include cytoplasmic translation, translation of selective oncogenic mRNAs, and stress granule (SG) formation (Fig. 6b, Supplementary Fig. 9) and may represent potential vulnerabilities for the development of therapeutic interventions. The seven RNA-binding proteins (RBPs) earlier identified through computational analysis as constituting an RBP-related prognostic signature in osteosarcoma (OS) include zinc finger CCCH-type containing antiviral 1 (ZC3HAV1), RNA-binding motif protein 34 (RBM34), and insulin-like growth factor 2 mRNA-binding protein-2 (IGF2BP2)[11]. Although our analysis did not reveal the same RBPs to be significantly enriched in the interactomes of OS, IGF2BP3, closely related to IGF2BP2, emerged as the most highly enriched RBP in the OS interactomes (Fig. 3d). Additionally, IGF2BP1 ranked among the top 20% of elevated RBPs in the OS interactomes. Similarly to ZC3HAV1, another zinc finger domain-containing RBP, ZCCHC3, was also found among the top 20% elevated RBPs. Several RBM proteins were observed to be either significantly elevated (RBM41, RBM47) or reduced (RBM14, RBMX, RBMS3, RBMS2) in the OS RNA interactomes. These findings imply that comparable groups of RBPs, such as IGF2BPs, zinc finger proteins, and RBMs, exhibit dysregulation across OS RNA interactomes, with specific RBPs within these categories demonstrating distinct alterations in the interactomes of various osteosarcomas.

The group of RBPs involved in SG formation includes the universal SG markers eIF4G1 and PABPC1, as well as the SG nucleators G3BP1, CAPRIN1, FMR1, DDX3X and PUM2 that are thought to mediate the condensation of stalled translation pre-initiation complexes into granules[70,71] together with non-nucleating RBPs. SG nucleators can drive spontaneous SG formation when overexpressed even in the absence of stress and impair SG assembly when downregulated. Further, YBX1 which we found to be systematically increased in all OS RNA interactomes also plays a critical role in SG formation by directly binding to and activating the translation of G3BP1 mRNAs[72]. SGs have emerged as a stress-adaptive strategy for cancer cells to enhance cell fitness and survival by coping with various stresses due to increased metabolic demand and overuse of nutrients[73]. SGs also contribute to cancer proliferation, metastasis, invasion and chemotherapy resistance[73]. SG nucleators have been reported to be upregulated in several tumor types when compared to normal tissue[74–76]. In sum, our data identify key components of stress granules to be enriched in OS

RNA interactomes suggesting that buffering of stress by facilitated formation of stress granules may contribute to the fitness of tumor cells in osteosarcoma.

Two additional signatures exclusively characterize the more aggressive OS. These are mitochondrial translation-related proteins and components of a Myc-centered network, respectively. (Figs. 6a, 8b). Myc is known to stimulate nuclear-encoded mitochondrial gene expression and to activate mitochondrial biogenesis, which is coupled to cell cycle progression[77]. Therefore, the enrichment of mitochondrial translation-related proteins in the OS RNA interactomes may at least in part result from the enhanced Myc-centered network in the more aggressive OS with high translation activity. Myc is also known to promote protein synthesis by stimulating ribosome biogenesis and the expression of key translation factors[78]. Consistent with this function, the more aggressive sarcomata with high levels of Myc expression were enriched for expression of genes related to ribosome biogenesis and other cellular functions related to RNA metabolism and translation (Fig. 6c, d). Functional analysis revealed enhanced protein synthesis of the clinically aggressive tumors compared to the less aggressive ones (Fig. 7a, c). Remarkably, the two aggressive osteosarcomata with Myc gene amplification and strongly elevated protein abundance showed higher vulnerability to translation inhibition (Figs. 7b, 7d). As a proof of concept, we used CHX for global inhibition of translation, revealing differential toxicity between the aggressive and the less aggressive osteosarcomata. Furthermore, we found the OS, and particularly the aggressive OS with highly active translation, to be especially susceptible to homoharringtonine (HHT), an inhibitor of translation FDA-approved for the treatment of patients with chronic myeloid leukemia who are resistant and/or intolerant to tyrosine kinase inhibitors[79]. These findings indicate a "translation-hungry" phenotype of the aggressive sarcomata, which underlies their therapeutic vulnerability to translation inhibitors. Interestingly, the GCTB, which also exhibits highly active protein synthesis, is not vulnerable to the translation inhibitors tested here, thereby indicating that the response to translation inhibition is not an unspecific feature of translationally active cells.

A particularly interesting group of differentially enriched RBPs showed changes in RNA binding without corresponding changes in protein abundance in the full proteome, which are exemplified by proteins involved in splicing (U2AF2, hnRNPs and RBM proteins) (Fig. 4, Supplementary Data 7). It is known that extensive post-translational modifications (PTM) of splicing factors are critical for mediating their highly dynamic function[80] and to regulate differential splicing in cancer. Notably, the RNA-binding domains of many RBPs contain residues that are frequent targets of PTMs[81] and PTMs have been directly shown to control RNA binding. Consistent with a functional role of this activity, dysregulation of PTMs of RBPs has been linked to the pathophysiology of various diseases including cancers[82]. Therefore, signaling pathways that regulate PTMs of these proteins potentially drive the differential RNA binding of this class of proteins in osteosarcoma and may represent targets for novel treatment strategies of osteosarcoma.

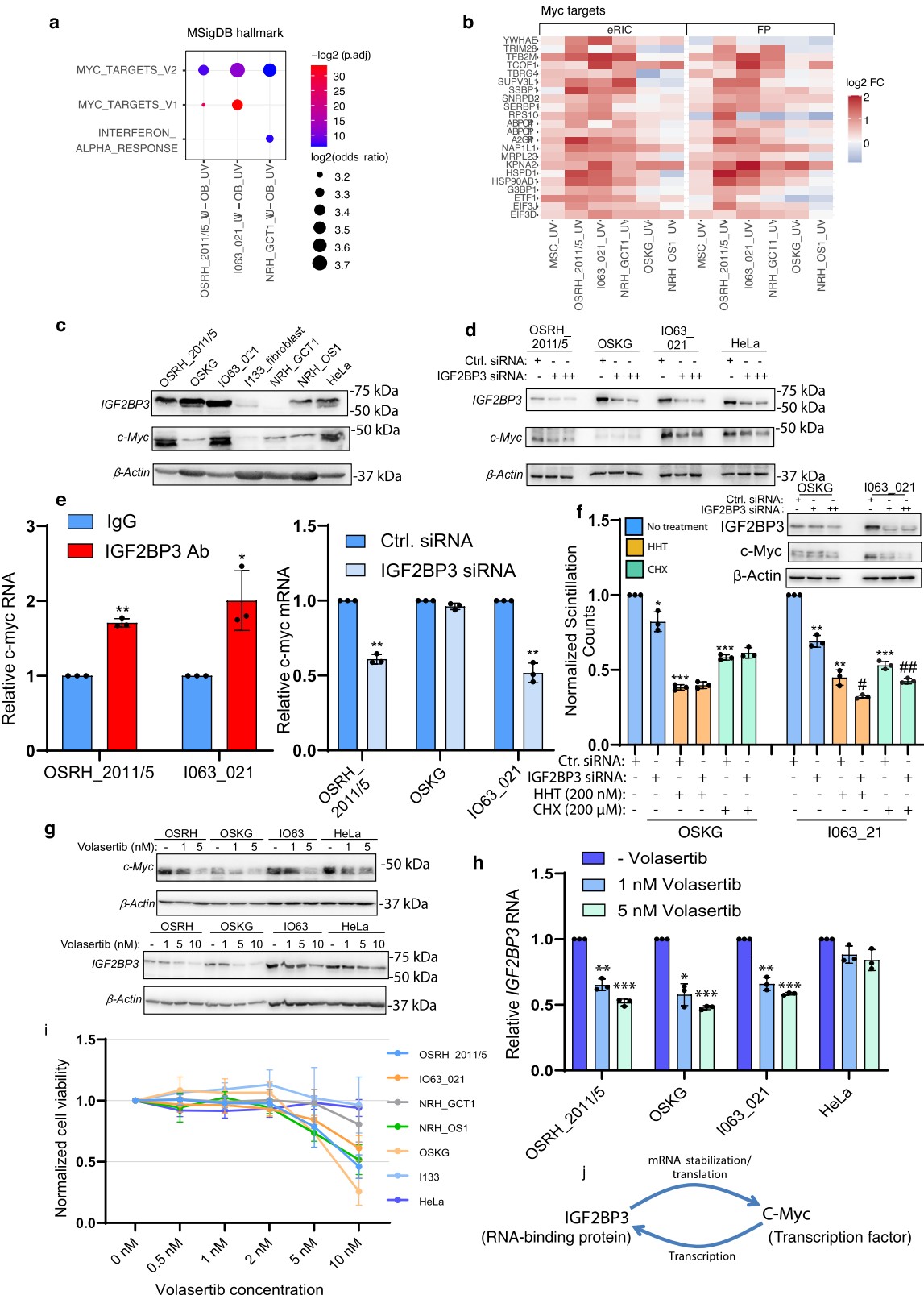

Finally, the data sets presented here include several RBPs that are altered in a tumor-specific manner. Amongst the top-ranked RBPs (20%) enriched in the OS/GCTB RNA interactomes (Fig.3d, left panel), we identified a set of RBPs that display particularly strong enrichment in individual tumors. These include PARP1 (7.7 fold in OSRH_2011/5), PTBP2 (6.5 fold in OSKG), and L1RE1 (10 fold in NRH_GCT1) (Supplementary Data 4). Such RBPs may thus represent

promising targets for personalized therapy. This potential is exemplified by PARP1 that represents the most abundant and founding member of the poly-ADP-ribosyltransferase (PARP) family, which catalyzes poly-ADP-ribosylation (PARylation). PARP1 was initially recognized for its crucial role in DNA repair, promoting PARylation at or near DNA damage sites, and the resulting poly(ADP-ribose) (PAR) chain serving as a docking platform for DNA repair proteins[83].

**Fig. 8 | An IGF2BP3-Myc positive feedback loop constitutes an oncogenic signature in OS with highly active translation. a** The altered RBPs in sarcoma RNA interactomes were subjected to the "hallmark" enrichment analysis using the "hallmark" gene sets from the MSigDB. Two sets of Myc targets are significantly enriched ($p$.adj<0.05) in the OS with highly active translation, OSRH_2011/5 and I063_021. The p values were obtained from the one-sided version of Fisher's exact test in the R package Clusterprofiler, after Benjamini-Hochberg adjustment. **b** RBPs that are Myc-targets were significantly enriched ($p$.adj<0.05) in the more aggressive sarcomata RNA interactomes compared to OB. The heatmap shows the enrichment level of each RBP in the RNA interactomes and the FP. The $p$ values were obtained from the one-sided version of Fisher's exact test in the R package Clusterprofiler, after Benjamini-Hochberg adjustment. **c** Representative Western blot showing IGF2BP3 and c-Myc protein abundance in OS cells, GCTB cells and in I133 fibroblasts and HeLa cells. The experiment was performed four times with similar results **d** Western blotting of lysates of fast-growing OS cells OSRH_2011/5 and I063_021 and slow-growing OS cells OSKG, transfected with 50 nM and 100 nM of IGF2BP3 siRNA or control siRNA using IGF2BP3, c-Myc and β-Actin antibodies. The experiment was repeated five times with similar results. **e** Quantitative RT-PCR of RNA, immunoprecipitated from lysates of OSRH_2011/5 and I063_021 cells with IGF2BP3 antibody and non-immune rabbit IgG, using c-Myc and β-Actin specific primers. The data represent fold change in *c-Myc* mRNA level in IGF2BP3 IP samples compared to IgG IP samples. The data represent mean ± SD derived from three biological replicates (left panel). * represents $p \leq 0.05$ (paired, two-tailed t-test, $p = 0.048$), ** represents $p \leq 0.005$ (paired, two-tailed t-test, $p = 0.002$). Quantitative RT-PCR of total RNA isolated from control siRNA or IGF2BP3 siRNA transfected cells using c-Myc and β-Actin specific primers. The data represent fold change in *c-Myc* mRNA level in IGF2BP3 siRNA-transfected cells compared to control siRNA-transfected cells. The data represent mean ± SD derived from three biological

replicates (right panel). ** represents $p \leq 0.01$ (paired, two-tailed t-test, $p = 0.002$, $p = 0.006$) **f** $^{35}$S metabolic labeling of I063_021 and OSKG cells transfected with 100 nM of siRNA against IGF2BP3 or control siRNA and treated with HHT (200 nM) or CHX (200 μM). The data represent mean ± SD derived from three biological replicates. * represents $p \leq 0.05$, ** represents $p \leq 0.01$ and *** represents $p \leq 0.001$ compared to control siRNA-transfected, drug-untreated cells (paired, two-tailed t-test, $p = 0.04$, p = 0.0003, $p = 0.0006$, $p = 0.005$, $p = 0.003$, $p = 0.0009$); # represents $p \leq 0.05$ and ## represents $p \leq 0.01$ compared to control siRNA-transfected, HHT-treated cells (paired, two-tailed t-test, $p = 0.04$, $p = 0.006$). Inset is a representative blot (from four independent experiments) of siRNA-mediated knockdown of IGF2BP3 using 50 nM and 100 nM siRNA. **g** Western blotting of OSRH_2011/5, I063_021, OSKG and HeLa cells either untreated or treated with 1 nM, 5 nM or 10 nM volasertib using c-Myc, IGF2BP3 and β-Actin antibodies. The experiment was repeated three times with similar results. **h** Quantitative RT-PCR of total RNA isolated from cells either untreated or treated with 1 nM and 5 nM volasertib using IGF2BP3 and β-Actin specific primers. The data represent fold change in *IGF2BP3* mRNA level in volasertib-treated cells compared to untreated cells. The data represent mean ± SD derived from three biological replicates. * represents $p \leq 0.05$ and ** represents $p \leq 0.01$ compared to volasertib untreated cells (paired, two-tailed t-test, $p = 0.05$, $p = 0.01$; $p = 0.01$, $p = 0.0003$; $p = 0.007$, p = 0.0002). **i** Cell viability, as measured by Celltitre-Glo assay, after treatment with volasertib (0–10 nM) for 24 h, normalized to untreated cells. Data represent mean ± SD derived from three biological replicates. **j** Schematic depiction of a positive feedback loop between IGF2BP3 and Myc, in which IGF2BP3 enhances the stabilization and translation of *c-Myc* mRNA while c-Myc enhances the transcription of *IGF2BP3* mRNA, resulting in increased expression of both proteins. Source data for graphs and blots are provided as a Source Data files.

Suppression of PARylation by PARP inhibitors (PARPi) has emerged as a promising therapeutic strategy for targeting tumor cells with homologous recombination repair (HR) deficiency[83]. In addition to its role in DNA repair, PARP1 has recently been revealed as a multifunctional RBP involved in several steps of RNA biogenesis and metabolism[84]. Interestingly, we previously noticed that the PARP-inhibitor talazoparib inhibits the clonogenic survival of OSRH_2011/5 cells significantly more profoundly than of OSKG[85]. This differential sensitivity could not be explained by PARP1-mediated DNA repair, because neither of these osteosarcomata appears to be HR deficient[85]. Our data may now explain this conundrum, since the RNA-related functions of PARP1 emerge as a candidate mechanism for the enhanced PARPi sensitivity of OSRH_2011/5, in which PARP1 RNA binding is strongly elevated. Thus, tumors with elevated RNA binding by PARP may respond to PARPi treatment even without HR deficiency, and the clinical development of a targeted approach deserves further attention in future studies.

Several individual RBPs identified here could also serve as therapeutic targets, especially YBX1, IGF2BP3 and MEX3A, which were most strongly enriched (up to 6.4 fold, 16.8 fold, and 7.4 fold, respectively) in all 4 OS RNA interactomes (Fig. 3d, Supplementary Data 4). Besides its DNA-binding function, YBX1 plays versatile RNA-dependent roles in pre-mRNA splicing, mRNA stabilization and translational regulation of targeted mRNAs affecting various pathways of cancer development, including the maintenance of stemness[36], SG formation by translational activation of the critical SG nucleator G3BP1[72], and stabilization of *Myc* transcripts[59,60], which are potential oncogenic features of the OS RNA interactome characterized here. Therefore, targeting YBX1 may lead to a multi-pronged interference with osteosarcoma biology.

We have focused our functional analyses on IGF2BP3, because it is known to modulate the biology of a variety of malignant tumors by regulating the stability and translation of oncogenes and tumor suppressor genes[28]. High level of IGF2BP3/IMP3 expression has been associated with metastatic OS in earlier studies, including in a microarray analysis comparing differentially expressed genes between metastatic and non-metastatic osteosarcoma cells[86,87]. Notably,

IGF2BP3 has been reported to stabilize *Myc* transcripts by cooperating with YBX1 in an N6-methyladenosine (m$^6$A)-dependent manner in myeloid leukemia cells[60]. At a different mechanistic level, Myc has been shown to bind to the promoter of the IGF2BP3 gene and enhance the transcription of its mRNA[66]. A positive feedback loop involving the stabilization of *N-Myc* RNA by IGF2BP3 and transcriptional upregulation of IGF2BP3 by MycN has been shown in neuroblastoma[88]. Here we have demonstrated that a positive feedback loop involving IGF2BP3 and Myc constitutes an oncogenic signature of the osteosarcomata, which show the highest enrichment of IGF2BP3 in their RNA interactomes (I063_021 and OSRH_2011/5) and are therapeutically vulnerable to the clinically approved Plk1 inhibitor volasertib. This positive feedback loop presents a particularly interesting drug target because of the multiplicity of RBPs involved in translation control, including IGF2BP3, regulated by Myc in these "translation-hungry" osteosarcoma cells (Fig. 8b). While such a positive feedback loop between IGF2BP3 as an RBP and c-Myc as a transcription factor has been shown for osteosarcoma in this study, this may be of wider interest in tumor biology and therapy as regulation of Myc by IGF2BP3 and vice versa has been attested in multiple tumor types[37], and a positive feedback loop between IGF2BP3 and MycN promoting the proliferation of neuroblastoma cells has also been demonstrated[88]. Such feedback loops between RBPs and transcription factors are particularly valuable in developing therapeutic strategies in cases when oncogenic transcription factors present difficult drug targets. It is also remarkable that the expression of Myc is low and of IGF2BP3 is undetectable in the sample of malignant giant cell tumor of bone, which shows high translation activity similar to the aggressive osteosarcomata but is not susceptible to the translation inhibitors tested in this study. These findings demonstrate that the effect of the translation inhibitors is not an unspecific response in translationally active tumor cells. By contrast, these findings indicate that the absence of the IGF2BP3-Myc positive feedback loop in this cell type, and other possible differences in post-transcriptional regulatory programs, may contribute to the differential response of this aggressive bone tumor to translation inhibitors.

Taken together, we have gained fundamental insights into the role of RBPs in the biology of osteosarcoma. Our findings offer rationales

for the development of RNA biology-based approaches for the treatment of this highly malignant bone tumor.

## Methods

### Ethics

The research described in this study follows all relevant ethical regulations and has been conducted following the approval by the ethics committee of the Medical Faculty of Heidelberg University, Germany and Committee for Ethics of Southeastern Norway.

### Patients

Primary tumor cells were derived from 4 osteosarcoma patients OSRH_2011/5, OSKG, I063_021 and NRH_OS1 and from one patient of malignant giant cell tumor of the bone (mGCTB), NRH_GCT1, following informed written consent of the patients and their guardians and approval of the ethics committee of the Medical Faculty of Heidelberg University, Germany (for samples OSRH, OSKG and I063_021) and the Committee for Ethics of Southeastern Norway (for samples NRH_OS1 and NRH_OS1). The material of the 5 patients analyzed here were included based on diagnosis and availability of suitable material. Of the 5 patients 1 was female and 4 were male. At the time of diagnosis the 5 patients analyzed here were 12-, 14-, 14-, 24- and 31-years old. All of these tumors were initially localized and classified as high-grade conventional osteosarcoma by histopathology. All patients received standard chemotherapy and resection of the primary tumor. OSRH_2011/5 and OSKG responded well to the preoperative chemotherapy (regression grades 2 and 3, respectively, according to Saltzer/Kuntschik). Patient NRH_OS1 had a poor response. OSRH_2011/5, I063_021 and NRH_OS1 suffered a relapse 24, 3, and 10 months after the initial diagnosis of primary disease, respectively. Both, OSRH_2011/5 and I063_021 had multifocal metastases including bone and lung, mediastinum (OSRH_2011/5) and liver (I063_021). Patient NRH_OS1 developed lung metastasis only. Patient OSKG did not develop relapse and remained in continuous complete remissions for more than 9 years of follow-up. In patient OSRH_2011/5 the relapse was rapidly progressive and the patient in such poor general condition that systemic treatment was not indicated. At the time of relapse, patient NRH_OS1 underwent pulmonary metastasectomy and then received treatment with high dose ifosfamide and mifamurtide resulting in stable disease. However, the patient later progressed and no further systemic treatment was given according to patient preference. Patients OSRH_2011/5 and I063_021 died of metastatic osteosarcoma 2 months after the diagnosis of relapse. Patient NRH_OS1 died 12 months after the diagnosis of relapse. In patient NRH_GCT1, malignant giant cell tumor of the bone (mGCTB) was diagnosed 10 years after surgery for benign GCT. As is characteristic for this entity, the disease behaved very aggressively and the patient died with metastatic disease only 5 months after the diagnosis of the malignant transformation.

### Cells and cell culture

The OSRH_2011/5 cells were obtained from the tumor tissue grown in an orthotopic xenotransplanted mouse model from the relapsed tumor as previously described[16]. The OSKG and I063_021 cells were generated directly from the clinical biopsies. The NRH_GCT1 and NRH_OS1 were generated from tumor tissues grown in xenotransplanted mouse models. The osteoblasts (OB, C-12720, PromoCell) and bone marrow-derived mesenchymal stem cells (MSCs, C-12974, PromoCell) were purchased from PromoCell, Germany. Cells of OSRH_2011/5, OSKG, I063_021, and NRH_GCT1 were cultured in DMEM medium (Thermo Fisher Scientific) supplemented with 10% fetal bovine serum (FBS, Thermo Fisher Scientific) and 1% nonessential amino acids (NEAA, Thermo Fisher Scientific). NRH_OS1 cells were cultured with addition of 0.2% Insulin-Transferrin-Ethanolamine-

Selenium (ITES, #17-839Z, Lonza) and 1% vitamin mix (#13-607 C, Lonza) to support the growth. Note that NRH_OS1 cells did not grow in the medium used for the other cells studied here. The OB and MSCs were cultured in Osteoblast Growth Medium (C-27001, PromoCell) and Mesenchymal Stem Cell Growth Medium (C-28009, PromoCell), respectively.

### DNA methylation and copy number variation profiling

The patient-derived cells from tumors of the 4 OS patients and the mGCTB patient were subjected to sarcoma classification based on DNA methylation data. All cells derived from the 4 OS tumors were predicted as high-grade osteosarcoma, and the cells derived from mGCTB (NRH_GCT1) were predicted as giant cell tumor of bone by the sarcoma classifier[89]. The analysis of copy number variation of the OS cells was performed according to Koelsche et al.[89]. Briefly, the copy number variations of genomic segments were inferred from the methylation array (Infinium MethylationEPICBeadChip microarray) data based on the R-package conumee after additional baseline correction (https://github.com/dstichel/conumee).

### Coupling of LNA oligonucleotides to magnetic beads

The coupling of locked nucleic acid (LNA) oligonucleotides to magnetic beads was performed as described before[14]. Custom-designed LNA oligonucleotides were synthesized with the sequence /5AmMC6/ +TT + TT + TT + TT + TT + TT + TT + TT + TT + TT ( + T: LNA thymidine, T: DNA thymidine). LNA oligonucleotides were dissolved in nuclease-free water to a final concentration of 100 μM. To prepare the carboxylated M-PVA C11 magnetic beads (#CMG- 203, PerkinElmer), the beads were washed 3 times with 5 volumes of 50 mM MES pH 6, resuspended in the original volume with MES buffer pH 6, and divided to 2 ml DNA LoBind tubes (Eppendorf) with 200 μl each. Fresh N-(3-Dimethylaminopropyl)-N'-ethylcarbodiimide hydrochloride (EDC, #E7750, Merck) solution were prepared in MES buffer pH 6 at 20 mg/ml. EDC-LNA solution was then prepared by adding every 200 μl of 100 μM LNA oligos to 1 ml of EDC solution. For coupling, 200 μl of washed and magnetized beads were resuspended in 1.2 ml of EDC-LNA solution in every 2 ml DNA LoBind tubes. The EDC-LNA-beads mixture was incubated at 50 °C for 5 h at 800 rpm with occasional pelleting and vortexing. After coupling, the beads in each 2 ml tube were washed twice with 1.5 ml PBS followed by inactivation of uncoupled carboxyl residues by incubating in 1.2 ml of 200 mM ethanolamine pH 8.5 (E9508, Merck) at 37 °C for 1 h at 800 rpm. After 3 times washes with 1.5 ml 1 M NaCl, the beads were stored in 200 μl of 0.1% PBS-Tween at 4 °C.

### Preparation of cell lysates for eRIC

Cells were grown on 500 cm² square dishes (#166508, Thermo Fisher Scientific) to reach 70-80% confluence. The cells were then washed twice with ice-cold PBS and subjected to UV crosslinking (UV) at 254 nm (150 mJ/cm²) in a Stralinker UV Crosslinker (Stratagene) or without crosslinking (noUV) as negative controls. The cells in each dish were scraped in 2 ml of lysis buffer containing 20 mM Tris-HCl pH 7.5, 500 mM LiCl, 1 mM EDTA, 5 mM dithiothreitol (DTT), 0.5% (w/v) LiDS, cOmplete protease inhibitor cocktail (#04693132001, Merck) on ice. The cell lysates were collected into 7 ml Precellys tubes (#432-0353, VWR) containing 500 μl 1 mm zirconia/silica beads (#11079110Z, BioSpec Products), followed by homogenization in a Precellys 24 Tissue Homogenizer (Bertin Instruments) at 5000 rpm for 10-20 sec at 4 °C until the cell lysate turning to not viscous anymore. For collection of the cell lysates, the 7 ml Precellys tubes containing cell lysates were punctured at the bottom using a hot needle, placed on a holder tube and centrifuged at 3000 rpm for 10 min at 4 °C. The cell lysates were then collected into fresh 15 ml DNA loBind tubes (Eppendorf) and snap frozen in liquid nitrogen and kept at −80 °C.

## Capture of RBPs using eRIC

The RNA interactome capture using eRIC was performed essentially as previously described[14,19], with minor adaptations. For each sample, 20 mg of input from both UV and noUV conditions were denatured at 60 °C for 10 min, quickly cooled down, and supplemented with fresh DTT to a final concentration of 5 mM. The cell lysates were then incubated with 600 µl of LNA-coupled beads equilibrated 3 times with 3 volumes of lysis buffer mentioned above. After incubation at 4 °C for 1 h with gentle rotation, the beads were magnetized and washed. Washes were done with 1 wash with lysis buffer (mentioned above), 2 washes with buffers 1 (20 mM Tris-HCL pH 7.5, 500 mM LiCl, 1 mM EDTA, 5 mM DTT, 0.1% (w/v) LiDS), 2 washes with buffer 2 (20 mM Tris-HCL pH 7.5, 500 mM LiCl, 1 mM EDTA, 5 mM DTT, 0.02% (v/v) NP-40) and 2 washes with buffer 3 (20 mM Tris-HCL pH 7.5, 200 mM LiCl, 1 mM EDTA, 5 mM DTT, 0.02% (v/v) NP-40) for 5 min of each wash at room temperature with gentle rotation. For the final wash, the beads were resuspended in 400 µl nuclease-free water and incubated for 5 min at 40 °C with 800 rpm rotation. Then 10% (40 µl) of the beads were eluted in 40 µl nuclease-free water using heat (90 °C) for 10 min with 800 rpm rotation, and the remaining beads were eluted in 300 µl RNase buffer (10 mM Tris-HCL pH 7.5, 150 mM NaCl, 5 mM DTT, 0.01% NP-40, 0.5 µl RNase A (#EN0531, Thermo Fisher Scientific), and 0.4 µl RNase T1 (#R1003, Merck)) for 1 h at 37 °C at 800 rpm. The heat eluates and RNase eluates were collected in new 1.5 ml DNA loBind tubes (Eppendorf). The heat eluates were used for RNA quantification and quality control via Nanodrop and bioanalyzer. The RNase eluates were supplemented with 2 µl 10% SDS and concentrated to 100 µl using a SpeedVac and subjected to protein quality control using SDS-PAGE followed by silver staining and western blot, and MS analyses for identification of captured proteins.

## Silver staining and western blot

For the experimental validation of eRIC on the protein level, the cell lysates (inputs) and the RNase eluates were analyzed by silver staining and western blot. For silver staining, 200 ng inputs and 8% of the RNase eluates were loaded on a 4 to 12% Bis-Tris gel (Thermo Fischer Scientific), followed by silver staining using the SilverQuest™ Silver Staining Kit (#LC6070, Thermo Fisher Scientific) according to the manufacturer's protocol. For western blot, 10 µg of inputs and 8% of the RNase eluates were separated on a 4 to 12% Bis-Tris gel and the proteins were transferred to a PVDF membrane. Followed by immunodetection using primary antibodies and horseradish peroxidase (HRP) conjugated secondary antibodies, the signal was developed using chemiluminescent detection (Western Lightning Plus ECL, #NEL104001EA, PerkinElmer) and visualized using a Fusion-FX7 Spectra imaging platform (VilberLourmat). Antibodies used: CSDE1 (#13319-1-AP, Proteintech, 1:4000 dilution), HuR (#11910-1-AP, Proteintech, 1:5000 dilution), hnRNPK (#11426-1-AP, Proteintech, 1:5000 dilution), Histone H3 (#9715 S, Cell Signaling Technology, 1:5000 dilution), α-tubulin (#T5168, Merck, 1:4000 dilution), β-actin (#A1978, Merck, 1:4000 dilution), C-Myc (#10828-1-AP, Proteintech, 1:3000 dilution), IGF2BP3 (#14642-1-AP, Proteintech, 1:5000 dilution), anti-mouse (#A9044, Merck, 1:10000), anti-rabbit (#A0545, Merck, 1:10000 dilution).

## RNA analysis on a bioanalyzer

Total RNA from eRIC cell lysates (inputs) was extracted using the RNA Clean & Concentrator-5 kit (#R1016, Zymo Research), followed by Turbo DNase (#AM2238, Thermo Fischer Scientific) treatment to remove DNA, and purified again using the RNA Clean & Concentrator-5 kit. The concentration of RNA purified from inputs and eRIC heat eluates were determined using Nanodrop 2000 (Thermo Fisher Scientific). For the sample loading, the RNA aliquots from inputs and eRIC heat eluates were diluted to 5 ng/µl, and 1 µl of each sample was analyzed using the RNA 6000 Pico Kit (#5067-1513,

Agilent) on an Agilent 2100 Bioanalyzer according to the manufacturer's instruction.

## MS sample preparation and TMT labeling

Proteins from inputs and RNase eluates of eRIC were incubated with DTT (10 mM in 50 mM HEPES pH 8.5) for 30 min at 56 °C for the reduction of disulfide bridges in cysteines. Reduced cysteines were alkylated with 2-chloroacetamide (20 mM in 50 mM HEPES pH 8.5) at room temperature for 30 min in the dark. Samples were further processed using the SP3 protocol[90,91] and digested with trypsin (sequencing grade, Promega) using an enzyme-to-protein ratio of 1:50 for overnight at 37 °C. Peptides were then recovered by collecting supernatant on a magnet and combining with a second elution wash of beads with HEPES buffer. Subsequently, peptides were labeled with TMT16plex Isobaric Label Reagent (#A44521, Thermo Fischer Scientific) according to the manufacturer's instructions. Samples were combined for the TMT16plex and further cleaned up using an OASIS® HLB µElution Plate (Waters). Offline high pH reverse phase fractionation was carried out on an Agilent 1200 Infinity high-performance liquid chromatography system, equipped with a Gemini C18 column (3 µm, 110 Å, 100 ×1.0 mm, Phenomenex)[92].

## LC-MS/MS

Peptides were analyzed on an UltiMate 3000 RSLC nano-LC system (Dionex) fitted with a trapping cartridge (µ-Precolumn C18 PepMap 100, 5 µm, 300 µm i.d. x 5 mm, 100 Å) and an analytical column (nanoEase™ M/Z HSS T3 column 75 µm x 250 mm C18, 1.8 µm, 100 Å, Waters). Trapping was carried out with a constant flow of trapping solution (0.05% trifluoroacetic acid in water) at 30 µl/min onto the trapping column for 6 min. Subsequently, peptides were eluted via the analytical column running solvent A (0.1% formic acid in water, 3% DMSO) with a constant flow of 0.3 µl/min, with increasing percentage of solvent B (0.1% formic acid in acetonitrile, 3% DMSO). The outlet of the analytical column was coupled directly to an Orbitrap Fusion™ Lumos™ Tribrid™ Mass Spectrometer (Thermo Fischer Scientific) using the Nanospray Flex™ ion source in positive ion mode.

The peptides were introduced into the Fusion Lumos via a Pico-Tip Emitter 360 µm OD x 20 µm ID; 10 µm tip (New Objective or CoAnn Technologies) and an applied spray voltage of 2.4 kV. The capillary temperature was set at 275 °C. Full mass scan was acquired with mass range 375–1500 m/z in profile mode in the orbitrap with resolution of 120,000. The filling time was set at maximum of 50 ms with an AGC target set to standard. Data-dependent acquisition (DDA) was performed with the resolution of the Orbitrap set to 30000, with a fill time of 94 ms and a limitation of $1\times10^5$ ions. A normalized collision energy of 34 was applied. MS2 data was acquired in profile mode.

## MS data analysis

The acquired MS data was processed using IsobarQuant[93] and Mascot (v2.2.07). Data were searched against the human Uniprot proteome database (UP000005640) along with common contaminants and reversed sequences. The following modifications were included into the search parameters: Carbamidomethyl (C) and TMT16 (K) (fixed modification), Acetyl (Protein N-term), Oxidation (M) and TMT16 (N-term) (variable modifications). For the full scan (MS1) a mass error tolerance of 10 ppm and for MS/MS (MS2) spectra of 0.02 Da was set. Further parameters include: trypsin as protease with an allowance of maximum two missed cleavages; a minimum peptide length of seven amino acids; at least two unique peptides were required for a protein identification. The false discovery rate on peptide and protein level was set to 0.01.

The raw output files (protein.txt) of IsobarQuant were processed using the R programming language (ISBN 3-900051-07-0). Only proteins that were quantified with at least two unique peptides were considered for the analysis. Moreover, only proteins which were

identified in both replicates were kept. Raw TMT reporter ion intensities ('signal_sum' columns) were first cleaned for batch effects using the limma package[94] and further normalized using the variance stabilization normalization (vsn) method[95]. Different normalization coefficients were estimated for UV crosslinked and non-crosslinked (noUV) samples to maintain the abundance difference. Missing values were imputed with 'knn' method using the Msnbase package[96]. Proteins were tested for differential abundance using the limma package. The replicate information was added as a factor in the design matrix given as an argument to the 'lmFit' function of limma. Imputed values were given a weight of 0.05 in the 'lmFit' function. The statistical significance was determined with a false discovery rate (FDR). For the UV versus noUV comparison, a protein with an FDR < 0.05 and a fold change (FC) ≥ 2 was annotated as an enriched hit (high confident RBP). For the comparative RNA interactome analyses (UV versus UV sample comparison from eRIC), a hit (FDR < 0.05 and FC ≥ 1.5) was defined for a protein with differential abundance in the two samples under comparison.

## Transcriptome sequencing and analysis

Total RNA was isolated from OS, GCTB, OB and MSC cells using TRIzol and cleaned up by RNA Clean and Concentrator kit-5 (Zymo Research, Cat. 1016). The quality and quantity of the RNA was estimated using BioAnalyzer (Agilent 2100) and Qubit 2.0 (Thermo Fisher) and 1 μg of each RNA sample was submitted for library preparation and sequencing. Samples were prepared following the NEB Next stranded RNA with the Poly(A) mRNA magnetic isolation module (NEB #E7490). Samples were standardized to 160 ng input, with 15 min fragmentation, no size selection, 1:30 adapter dilution, and 13 PCR cycles. Libraries were pooled equimolarly and size selected with a 0.6x bead ratio. Samples were sequenced on the Illumina NextSeq 2000 platform with a 100-paired end sequencing program on a P2 flowcell. The raw reads were trimmed using Cutadapt (v4.4)[97] and aligned to the human genome (GRCh38.p13) with STAR (v2.7.10.b)[98]. Aligned reads were summarized with featureCounts (v2.0.6)[99]. DESeq2 (v1.40.2)[100] using local dispersion fit and the Wald test with IHW[101] for multiple hypothesis correction was used to determine significantly differentially regulated genes in each test sample versus control samples comparison, respectively (adjusted $p$ value < 0.05; absolute log2 fold change > 1). ClusterProfiler[102] was used to perform the Gene Ontology analysis for significantly enriched genes (adjusted $p$ value < 0.05; log2 fold change > 1).

## Gene Ontology (GO) enrichment analysis

The GO enrichment analysis of the bone-cell RNA interactome (Fig. 2d) and the proteins from the 7 clusters derived from the comparative RNA interactome analysis (Fig. 5d) were performed using the R package clusterProfiler[102] and the annotation package org.Hs.eg.db. The thresholds for enrichment were set to default: Benjamini-Hochberg multiple testing corrected $p$ value cutoff of 0.05 ($p$.adj<0.05) and $q$ value cutoff of 0.2 ($q$ < 0.2). Proteins identified in the full proteome (6134 proteins) were used as background.

## Characterization of the bone/mesenchymal-cell RNA interactome

To compare the here identified bone/mesenchymal-cell RNA interactome with previously identified poly(A) binding RBPs, human RNA interactome studies (RIC and eRIC) from other cell types (HEK293, HuH7, HeLa, K562, Jurkat, nuclear and cytoplasmic fraction of HuH7) were obtained from the RBPbase (https://rbpbase.shiny.embl.de, v.0.2.0). In addition, a list of previously known RBPs along with RBPs with known RBD obtained from Backlund et al.[19] were also included for comparison. The 33 proteins uniquely identified in the bone/mesenchymal-cell RNA interactome were considered as potential bone cell-specific RBPs (Supplementary Fig. 2, Supplementary Data 3). The

(metabolic) enzymatic function annotation of the bone/mesenchyma-cell RNA interactome in Fig. 2e was done according to the list of (metabolic) enzymes from the RBPbase. Protein domain enrichment analysis of the bone/mesenchyma-cell RNA interactome in Fig. 2f was conducted using the Pfam database via DAVID platform (version 6.8)[103] with Benjamini corrected $p$-value ($p$.adj.<0.05). The analysis of the intrinsically disordered properties of proteins from bone/mesenchyma-cell RNA interactome in Fig. 2g was performed based on the DescribePROT database[104].

## Identification of altered RBPs in RNA interactomes

The pairwise comparison of RNA interactomes including each sarcoma (OS or GCTB) to OB or to MSC and MSC to OB were performed which resulted in 350 RBPs with significantly different abundance (FDR < 0.05, FC ≥ 1.5) in all the comparisons. The hierarchical clustering of these 350 RBPs were performed using the Manhattan distance with Ward.D2 clustering algorithm in Fig. 5a. To obtain the lists of proteins associated with "mitochondrial translation", "translation" and "cytoplasmic stress granule" which are GO terms enriched for the elevated proteins in the OS RNA interactome compared to OB, the GO terms of all elevated proteins in OS and GCTB were retrieved from the Uniprot database. The group of proteins with "mitochondria" and "stress granule" related annotation in GO terms under the "biological process" category was graphically represented in the heatmaps of Fig. 6a and Supplementary Fig. 9, respectively. The group of "cytoplasmic translation" related proteins were obtained by subtracting the "mitochondrial translation" related proteins from the "translation" related proteins under the "biological process" category and graphically represented in the heatmap of Fig. 6b.

## Consensus clustering

Consensus clustering of all the RNA interactomes studied was performed based on the protein abundances of UV samples using the R package ConsensusClusterPlus. It applied hierarchical clustering and subsampling 80% samples at each time and repeatedly for 100 times to achieve robust clustering. This determines "pairwise consensus values" which represents how frequently 2 samples fall into the same cluster, and therefore allows the assessment of the cluster stability and identification of optimal cluster number (k). For each k, a final agglomerative hierarchical clustering is generated based on sample distance (1- Pearson correlation coefficient).

## Assessment of global protein synthesis using metabolic labeling assay

Cells were seeded in six-well dishes at a density which allows growing to around 60-70% confluence the next day. The next day, 30 min prior $^{35}$S labeling, the depletion medium was added to the cells which replaced the complete DMEM with DMEM without methionine and cysteine (#21013024, Thermo Fisher Scientific) and the complete FBS with dialyzed FBS (#F0392, Merck) to deplete the internal stores of methionine and cysteine in cells. Other additives (NEAA, ITES and vitamin mix) were maintained at the same amount as in the normal culture conditions. Then 20 μCi/ml of [S35]Met-label (70% L-[S35] Methionine, -25% L-[S35] Cysteine, #SCIS-103, Hartmann Analytic) were added to the medium for 30 min to label newly synthesized proteins. After labeling, cells were washed 3 times with cold PBS and lysed in high-salt RIPA buffer (50 mM Tris/HCl pH 8.0, 500 mM NaCl, 1% NP-40, 0.1% SDS, 0.5% sodium deoxycholate, cOmplete protease inhibitor cocktail). The cell lysate was then collected, incubated for 10 min on ice, vortexed and centrifuged at 16,000 g for 10 min at 4 °C. The supernatant was transferred to a new tube and protein concentration was measured using a DC protein assay. For autoradiographic analysis, 10 μg cell lysate for each sample were separated on an SDS-PAGE gel, followed by coomassie staining for visualizing the proteins for equal loading. The gel was then dried on a piece of

Whatman filter paper under vacuum and heat (80 °C) for 2 h. The dried gel was exposed to an autoradiography film for 2 days and developed. For quantification of $^{35}$S incorporation by scintillation counting, 10 µl cell lysate were spotted onto a glass microfiber filter (Whatman) and air dried. After drying, proteins on the filter were precipitated with cold 15% TCA for 30 min and followed by one wash with cold 15% TCA and two washes with cold 100% ethanol. The filter was dried, immersed in scintillation solution and the radioactivity was measured using a scintillation counter (WallacWinSpectral, PerkinElmer). The scintillation counts were normalized to total protein amount. Cells were treated with cycloheximide (CHX, 50 µg/ml, #C7698, Sigma-Aldrich) or homoharringtonine (HHT, 5 mM in 75% DMSO, #SML1091 Sigma-Aldrich) for 2 h prior to $^{35}$S labeling.

### Cell proliferation and population doubling time
To calculate the proliferation rate, $1.3 \times 10^5$ of OSRH_2011/5 cells, $0.65 \times 10^5$ of OSKG cells, $0.05 \times 10^5$ of I063_021 cells, $0.7 \times 10^5$ of NRH_OS1, $0.3 \times 10^5$ of NRH_GCT1 and $0.2 \times 10^5$ of I133_fibroblasts cells were seeded on each well of 12-well plates. The viable cell numbers were determined by trypan blue staining and counting of the cells using a TC20 automated cell counter (Bio-Rad) every 24 h for a period of 5 days. The population doubling time of the exponential phase of growth of the cells were calculated using the online tool https://www.doubling-time.com/compute.php.

### Cell viability assay
To assess the cell viability upon CHX, homoharringtonine (HHT) and volasertib treatment, cells were seeded on 48-well or 96-well plates at a density which allows growing to 60-70% confluence the next day. The next day, cells were treated with freshly prepared CHX solutions at final concentrations of 100, 150, 200, 250, and 300 µg/ml for 5.5 h After treatment, cells were quickly washed twice with PBS to remove floating dead cells. Cells in each well were incubated with 200 µl cell culture medium containing 30 µl of Celltiter Blue (#G8080, Promega) at 37 °C for 3-4 h. The assay was stopped and stabilized by the addition of 100 µl of 3% SDS in each well. From each well, 100 µl of the reaction solution was taken into a 96-well plate and the fluorescence was recorded at 560/590 nm using a SpectraMax M2 microplate reader (Molecular Devices). Cells were treated with HHT at final concentrations of 50, 100, 200, 500 and 1000 nM for 24 h. Cell were treated with volasertib (56.56 mM in DMSO, #S2235 Selleckchem) at final concentrations of 0.5, 1, 2, 5 and 10 nM for 48 h. After treatment cells were washed with PBS and incubated with 20 µl Celltiter-Glo (#G7570, Promega) for 10 min, following which luminescence was measured in an Infinite® 200 PRO multimode plate reader (Tecan). Cell viability was calculated by subtracting the cell culture medium background and normalized to the no treatment control.

### siRNA transfection
Cells were transfected with 50 nM or 100 nM siGENOME SMARTpool siRNA against IGF2BP3 (D-003976, Dharmacon) with Lipofectamine RNAiMAX transfection reagent (#13778075, Thermo Fisher Scientific). 48 h after siRNA transfection cells were harvested and lysed for western blotting and RNA isolation or treated with homoharringtonine or cycloheximide for 2 h before $^{35}$S-Met/Cys metabolic labeling.

### Statistics and Reproducibility
The study was conducted with 5 patient-derived osteogenic cancer cell lines and two commercially obtained control cells lines. No statistical method was used to predetermine sample size. The RNA interactome capture, proteomic and transcriptomic data and cell doubling time data are obtained from two biological replicates. All other experimental results are from at least three independent biological replicates. No data were excluded from the analyses. The Investigators were not blinded to allocation during experiments and outcome assessment.

### Reporting summary
Further information on research design is available in the Nature Portfolio Reporting Summary linked to this article.

## Data availability
The mass spectrometry proteomics data have been deposited to the ProteomeXchange Consortium via the PRIDE[105] partner repository with the dataset identifier PXD038185. The RNAseq transcriptomic data have been deposited in NCBI's Gene Expression Omnibus[106] and are accessible through GEO Series accession number GSE246405. The transcriptomic, eCLIPseq and RIPseq datasets used to generate Supplementary Fig. 8 and Supplementary Data 8 are publicly available as supplementary data in Palanichamy et al.[52], Qiu et al.[53]. and in the ENCODE database (https://www.encodeproject.org/genes/8165/). The processed complete eRIC, full proteome and transcriptome data is available as Supplementary Data 9 and 10. The remaining data are available within the Article, Supplementary Information or Source Data files. Source data are provided with this paper.

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

## Acknowledgements

We thank Nicole Dickemann for her excellent technical assistance. We thank Manouk Gerritsen, Gabriele Neu-Yilik, Daria Lavysh, Michael Backlund and Jonas Becker for their fruitful discussions and help. M.W.H. and A.E.K. are supported by a grant from the Deutsche Forschungsgemeinschaft (DFG; KU 563/18-1). O.M. is supported by The Norwegian Cancer Society (#144385).

## Author contributions

Y.Z., A.E.K. and M.W.H. designed the project. Y.Z. and P.S.R. performed the experiments, data analyses and drafted the manuscript. M.R. performed the MS sample preparation and MS runs. J.Z. and F.S. performed the MS data analyses. T.S. and S.S. performed the RNAseq data analyses. J.I.P.-P contributed to the training of the eRIC protocol. C.F. and A.N.B contributed to coupling of the LNA-beads. E.K.R. provided the OSRH_2011/5 and OSKG cells and the clinical information of these two patients. O.M. and L.A.M.-Z. provided NRH_OS1 and NRH_GCT1 cells. K.B. provided clinical information of the patients NRH_OS1 and NRH_GCT1. M.N., C.B. and B.L. provided tumor tissue and the clinical information. A.v.D. performed DNA methylation and copy number variation data analyses and interpretation. M.W.H. and A.E.K. designed and coordinated the study and finalized the manuscript.

## Funding

## Competing interests

The authors declare no competing interests.

## Additional information

[1]Molecular Medicine Partnership Unit (MMPU), Heidelberg University and European Molecular Biology Laboratory (EMBL), Heidelberg, Germany. [2]Department of Pediatric Oncology, Hematology and Immunology, Heidelberg University Hospital, Heidelberg, Germany. [3]European Molecular Biology Laboratory (EMBL), Heidelberg, Germany. [4]Department of Clinical Science, University of Bergen, Bergen, Norway. [5]Department of Tumor Biology, Institute for Cancer Research, Oslo University Hospital, Oslo, Norway. [6]Genomics Core Facility, Department of Core Facilities, Institute for Cancer Research, Oslo University Hospital, Oslo, Norway. [7]Department of Neuropathology, Institute of Pathology, Heidelberg University Hospital, Heidelberg, Germany. [8]Clinical Cooperation Unit Neuropathology, German Cancer Research Center (DKFZ), German Cancer Consortium (DKTK), and Hopp Children's Cancer Center at the NCT Heidelberg (KiTZ), Heidelberg, Germany. [9]Department of Oncology, Oslo University Hospital, Oslo, Norway. [10]Department of Pediatrics and Children's Cancer Research Center, Technical University of Munich, School of Medicine, Munich, Germany. [11]Pediatric Hematology and Oncology, Klinikum Kassel, Kassel, Germany. [12]Department of Pediatric Oncology, Hematology and Immunology, Olga Hospital, Stuttgart, Germany. [13]Department of Orthopaedics, Trauma Surgery and Paraplegiology, Heidelberg University Hospital, Heidelberg, Germany. [14]Clinical Cooperation Unit Pediatric Leukemia, German Cancer Research Center (DKFZ) and Heidelberg University, Heidelberg, Germany. [15]These authors contributed equally: Yang Zhou, Partho Sarothi Ray. ✉e-mail: hentze@embl.org; andreas.kulozik@med.uni-heidelberg.de

