## [Peer Review File · Nature Communications]

Reviewers' Comments:

Reviewer #1:

Remarks to the Author:

The noteworthy results presented in this paper include the application of an innovative eRIC technology to determine an RNA-binding interactome for 4 patient-derived osteosarcoma specimens and 1 malignant GCT, comparing these to control cells representing mesenchymal stem cells and osteoblasts, as well to each other by subgroup analyses. The only prior report of RNA-binding proteins in osteosarcomas that I have encountered is PMC9091083, which did not directly interrogate protein interactomes experimentally, but used reported STRING database associations to impute interactomes based on differentially expressed RBPs. I would therefore conclude that this work is novel, but suffers from an extremely small sample size. I am afraid that beyond a description of these few cell samples, any further claims could not be confidently supported by so small a sample size of so heterogeneous a cancer type. If they had used this descriptive screening experiment as foundation for a variety of functional experiments to test specific hypotheses, that would make more sense, but they really provided only functional experiments with cyclohexamide.

Reviewer #2:

Remarks to the Author:

In this work, Zhou et al., have characterized the complement of RNA-binding proteins (RBPs) relevant to osteosarcoma using enhanced RNA Interactome Capture (eRIC) comparing, for the first time, the RNA interactomes of five malignant bone tumors (four primary patient-derived osteosarcomata (OS) and one malignant giant cell tumor of the bone) with normal mesenchymal stem cells and osteoblasts. I think this is the highlight of the work as there are not many studies (to my knowledge) on RBP profiling using cells derived from patients.

The data presented here reveal a limited rewiring of the RBPome in osteosarcoma that may contribute to OS tumorigenesis. These alterations are mainly related to cytosolic and mitochondrial translation, among other functions. These modified functions are consistent with previous work in other cancers, but also reflect the 'knowns' in the connections between tumors and RNA metabolism. No functional analysis of a novel discovery derived from this work is provided.

Therefore, although the work presents consistent and robust data and it is technically very well done, no further advances have been taken into the characterization of the RNA interactome data that will help to broaden the understanding of the molecular mechanisms governing the oncogenesis of OS.

Major points:

Page 7, line 143. FC 1.5 is in log 2 scale? In that case it will be 2.84 fold increase in the linear scale. As RBPs often work in a 'team', smaller changes can still be functionally relevant if reproducible. As the FDR accounts for amplitude and reproducibility of the effect, I am unsure of adding a fold change cut off is beneficial as may 'artificially' identify sample-specific proteins that are also altered in other samples but display a lower fold change and did not pass the 1.5 FC threshold, as shown later in Figure 3d. I think the comparison of fold changes between samples is key as can detect 'trends' that are not significant in (for example) slow growing cells but become significant in fast growing cells and vice versa. Therefore, Figure 3a and c can be very much affected by (over) arbitrary cut offs.

Figure 4. There are proteins such as CAPRIN1 (first two panels) and others that show correlation but are shown in group III. Note that the fold changes in RIC and whole cell proteome are not comparable in magnitude (as the complexity of the sample is different) but in 'direction'. In other words, it is possible to classify a protein as having consistent positive or negative change between RBPome and proteome, no change in one of the two (i.e. near to zero in the Y, X axis or both) or 'anti-correlation' (when opposite directions in the changes in RNA binding and abundance).

However, it is difficult to make assumptions regarding a protein with the same direction fold change but differences in magnitude, as happens for CAPRIN1, DHX30 and others in that area of the plot. I would though revisit the strategy for clustering.

Figure 5. Authors indicate that the aggressiveness of the OS correlates with the amount of the changes in the RNA interactome. However, it can also be that the cells become more different to the original OB as more aggressive the tumor is. In that scenario, the changes in RBP profile might just be a consequence of the differences in overall cell 'identity'. Therefore, similar analyses should be done for the total proteome and the transcriptome to determine whether the differences between samples are mostly restricted to RBPs or all widespread across a wide range of cellular pathways. If the second, which pathways are more affected.

Functional outputs. While good OMICS data can be the basis for hypothesis-driven work, I feel some degree of functional validation is required to differentiate between causes and consequences. I understand such work cannot be applied to all the discoveries of a proteome-wide study, but can at least be applied to an example to illustrate the value of the dataset. Particularly, when there is some novelty in a given discovery.

Conclusion: the work is, in principle, technically sound and original in the sense of employing patient-derived cells. On the other hand, evidence of the role of RBP rewiring in inducing OS aggressiveness is missing. Functional validation would strengthen the conclusions and test the value of the dataset.

Minor points:

Page 5 lane 112 - the analysis has still 5% FDR, which means that 5 out of 100 proteins will be wrongly assigned. Therefore, bona fide sounds too adventurous and I would rather use 'high confidence' RBP. It would be different if the protein has been reported as RBP by different studies. In such case, there will be an accumulated high probability for the protein to be an RBP.

Page 6, lane 134 - has not been known > rephrase. 'remains unknown'?

Figure 2b: Venn-diagram is missing.

Figure 3a: Bar plot is missing.

Figure 3a: "The number of elevated and reduced RBPs are indicated in blue and orange respectively". The numbers in the figure have no color code.

Reviewer #3:

Remarks to the Author:

This manuscript presents a characterization of the whole proteome and the RNA-binding proteome of osteosarcoma and bone tumor patient-derived cells, compared to normal mesenchymal stem cells and osteoblasts.

This characterization provides an interesting preliminary step in understanding the role of RNA-binding proteins in osteosarcoma, but further analyses and experiments would be essential to improve the significance of the results, link the observations to any pathogenic mechanism and identify promising therapeutic targets.

Specific points:

- In the introduction, the authors say: "In the current work, we comprehensively characterized RNA-RBP interactions in OS in an unbiased and systematic fashion by using enhanced RNA interactome capture (eRIC)". Yet the experiments are characterizing only the protein side, not the RNA side.

Parallel RNA-seq experiments would be crucial to understand whether the differential RNA-binding proteome is linked to alterations in RNA abundances and aberrant splicing of RNA isoforms (particularly relevant since the authors identify a group of splicing factors with altered RNA-binding). Although the technique will not capture individual RBP-RNA interactions, the characterization and comparison of altered protein binding sites on RNAs would be essential to

identify pathogenic post-transcriptional RNA processes mediated by RNA binding proteins. Focusing on the protein side of the interaction is only half of the story.

- The identification of several RBPs involved in Stress Granule formation is potentially interesting, but it should be validated by additional evidence and dedicated experiments: can the authors see alterations in stress granule formation in osteosarcoma cells by microscopy approaches? Are osteosarcoma cells more resistant to stress? Are osteosarcoma cells more vulnerable to stress granule inhibitors? How is stress granule formation connected with the increased vulnerability to translation inhibition observed by the authors?
- The authors show improved toxicity of cycloheximide in more aggressive osteosarcoma cells. As the authors say, this is only a proof of concept. Apart from exploring the mechanisms underlying alterations in cytoplasmic and mitochondrial translation, additional evidence with clinically feasible translation inhibitors (such as homoharringtonine, suggested by the authors) would be important to confirm the results and show that this vulnerability is therapeutically amenable.
- The work feels poorly connected to previous osteosarcoma literature. The authors in the abstract say: "Genomic, epigenomic and transcriptomic analyses have demonstrated the exquisite molecular complexity of this tumor, but haven't sufficiently defined the underlying mechanisms or identified promising therapeutic targets". How does this systematic analysis of RNA binding proteins relate to previous characterizations of osteosarcoma by omics approaches? A comparative analysis would improve the significance of the results by linking the observations presented in the manuscript to previous omics characterizations of osteosarcoma.
- Could differences in sample composition or cell heterogeneity act as a possible bias when comparing osteosarcoma samples with normal mesenchymal stem cells and osteoblasts, as well as when comparing different tumor samples?
- Is there any link between altered RNA-binding proteins and genomic lesions, copy number alterations and DNA methylation changes observed by the authors in osteosarcoma cells (Supplementary Figure 1)?
- PCA/MDS plots of samples based on total proteomics and RNA-binding proteomics measurements would be important to capture and visualize overall similarities/differences among total protein and RNA interactome samples between tumor cells, normal mesenchymal stem cells and osteoblasts.
- Some of the figures (e.g. 2B or 3A) had partially missing graphical elements in the manuscript pdf file. For example, in Figure 3A, I could only see numbers, but not bars. I would check for possible compatibility issues due to different operating system software.

I thank the authors for the hard work on this.

Point-by-point response to reviewers' comments:

Reviewer #1 - Expertise in osteosarcoma, preclinical models - (Remarks to the Author):

The noteworthy results presented in this paper include the application of an innovative eRIC technology to determine an RNA-binding interactome for 4 patient-derived osteosarcoma specimens and 1 malignant GCT, comparing these to control cells representing mesenchymal stem cells and osteoblasts, as well to each other by subgroup analyses. The only prior report of RNA-binding proteins in osteosarcomas that I have encountered is PMC9091083, which did not directly interrogate protein interactomes experimentally, but used reported STRING database associations to impute interactomes based on differentially expressed RBPs. I would therefore conclude that this work is novel, but suffers from an extremely small sample size. I am afraid that beyond a description of these few cell samples, any further claims could not be confidently supported by so small a sample size of so heterogeneous a cancer type. If they had used this descriptive screening experiment as foundation for a variety of functional experiments to test specific hypotheses, that would make more sense, but they really provided only functional experiments with cyclohexamide.

Response: We appreciate the reviewer's acknowledgment of the novelty of our work, particularly the identification of a lack of prior experimental studies investigating RBP interactomes in osteosarcomas. We concur with the reviewer's observation that our sample size is limited. This limitation arises because we use patient-derived cells and the eRIC protocol is work- and resource-intensive while yielding unique insights. We think that this work represents crucial "proof of concept" linking differential RBP interactomes in patient-derived osteosarcoma (OS) and giant cell tumor (GCT) cells with phenotypic variations, particularly differential translation levels, which underlie potential therapeutic vulnerabilities. In response to the reviewer's valuable suggestion, we have expanded our functional investigations into translation inhibition, with a specific focus on the translational potential of our findings. To this end, we utilized homoharringtonine (HHT), a translation inhibitor approved for clinical use in treating chronic myeloid leukemia (CML). Our new data demonstrate that HHT treatment exerts a differential effect on the patient-derived osteosarcoma cells analyzed in this study. Notably, the inhibitory effect of HHT, at therapeutic plasma concentrations, is more pronounced in the "translation-hungry" OS cells compared to cells with lower rates of protein synthesis (Fig. 7). Importantly, our new findings reveal that this vulnerability to translation inhibition is not merely a function of growth rate, as the equally rapidly proliferating malignant GCT cells do not exhibit susceptibility to HHT treatment.

Furthermore, we have extended our functional investigations by conducting new analyses of the role of IGF2BP3, which we identified as the most highly enriched RBP in osteosarcoma (Fig. 8). Specifically, we demonstrate that elevated IGF2BP3 levels are positively correlated with both, high c-Myc protein and mRNA levels. Knockdown of IGF2BP3 results in a reduction of c-Myc expression, even in the context of c-Myc chromosomal amplification, indicating post-transcriptional regulation of c-Myc expression by IGF2BP3. Additionally, knockdown of IGF2BP3 leads to reduced translation in one of the osteosarcomata characterized by high translation rates. Conversely, the inhibition of c-Myc by volasertib, a clinically approved Plk1 inhibitor known to inhibit c-Myc, also reduces IGF2BP3 expression. In line with the role of this positive feedback loop in the viability of osteosarcoma cells, volasertib exerts its most significant

impact on OS cells characterized by the highest levels of IGF2BP3. Collectively, our new data highlight a positive feedback loop regulating the differential expression of the RNA-binding protein IGF2BP3 and the potent tumor-promoting gene c-Myc in osteosarcoma. Importantly, our findings reveal that this positive feedback loop is susceptible to clinically approved drugs, thus underscoring the translational potential of our novel discoveries.

Reviewer #2 - Expertise in RBPs, RIC, mass-spec (Remarks to the Author):

In this work, Zhou et al., have characterized the complement of RNA-binding proteins (RBPs) relevant to osteosarcoma using enhanced RNA Interactome Capture (eRIC) comparing, for the first time, the RNA interactomes of five malignant bone tumors (four primary patient-derived osteosarcomata (OS) and one malignant giant cell tumor of the bone) with normal mesenchymal stem cells and osteoblasts. I think this is the highlight of the work as there are not many studies (to my knowledge) on RBP profiling using cells derived from patients.

The data presented here reveal a limited rewiring of the RBPome in osteosarcoma that may contribute to OS tumorigenesis. These alterations are mainly related to cytosolic and mitochondrial translation, among other functions. These modified functions are consistent with previous work in other cancers, but also reflect the 'knowns' in the connections between tumors and RNA metabolism. No functional analysis of a novel discovery derived from this work is provided.

Therefore, although the work presents consistent and robust data and it is technically very well done, no further advances have been taken into the characterization of the RNA interactome data that will help to broad the understanding of the molecular mechanisms governing the oncogenesis of OS.

Response: We are grateful for the reviewer's positive feedback concerning the novelty, significance, and technical rigor of our work. In response to the reviewer's comments, we have substantially enhanced the characterization of the RNA interactome data and followed-up on specific mechanistic insights derived from the RBP interactome analysis. Below, we provide detailed responses to the points raised by the reviewer:

Major points:

Page 7, lane 143. FC 1.5 is in log 2 scale? In that case it will be 2.84 fold increase in the linear scale. As RBPs often work in a 'team', smaller changes can still be functionally relevant if reproducible. As the FDR accounts for amplitude and reproducibility of the effect, I am unsure of adding a fold change cut off is beneficial as may 'artificially' identifies sample-specific proteins that are also altered in other samples but display a lower fold change and did not pass the 1.5 FC threshold, as shown later in Figure 3d. I think the comparison of fold changes between samples is key as can detect 'trends' that are not significant in (for example) slow growing cells but become significant in fast growing cells and vice versa. Therefore, Figure 3a and c can be very much affected by (over) arbitrary cut offs.

Response: We appreciate highlighting an ambiguity present in the previous version of our manuscript. As elucidated on Page 7, line 143, the notation "FC 1.5" represents values in a linear scale, although in the Excel files (supplementary data 4) the corresponding log₂ value (log₂ FC = 0.6) has been used. This has now been explicitly stated in the text. The deliberate utilization of this low cut-off value aligns with the rationale elaborated upon by the reviewer. This lower cut-off threshold has enabled us to conduct comparative analyses of differentially altered RBPs across various cells, as depicted in Figure 5a. Consequently, it has facilitated the identification of both shared patterns and distinctions between the RNA interactomes of these patient-derived cells.

Figure 4. There are proteins such as CAPRIN1 (first two panels) and others that show correlation but are showed in group III. Note that the fold changes in RIC and whole cell proteome are not comparable in magnitude (as the complexity of the sample is different) but in 'direction'. In other words, it is possible to classify a protein as having consistent positive or negative change between RBPome and proteome, no change in one of the two (i.e. near to zero in the Y, X axis or both) or 'anti-correlation' (when opposite directions in the changes in RNA binding an abundance). However, it is difficult to make assumptions regarding a protein with the same direction fold change but differences in magnitude, as happens for CAPRIN1, DHX30 and others in that area of the plot. I would though revisit the strategy for clustering.

Response: We acknowledge the fact that the correlatable changes of protein in the eRIC and the full proteome are in "direction" and not in the "magnitude" of the changes because of the differences in the complexity of the samples. In order to define a threshold for the changes, we have indicated groups of proteins in the scatterplots in Fig 4 which show a log₂ FC ≥ 0.6 (linear FC ≥ 1.5) and FDR < 0.5 in the full proteome or eRIC or both. Any protein, such as CAPRIN1 and DHX30, which is changed above this threshold, has been indicated by the colour-coded groups. We agree that this is a subjective strategy for the clustering but have retained it as it provides at least a partially quantitative method for grouping the proteins in this complex comparison.

Figure 5. Authors indicate that the aggressiveness of the OS correlates with the amount of the changes in the RNA interactome. However, it can also be that the cells become more different to the original OB as more aggressive the tumor is. In that scenario, the changes in RBP profile might just be a consequence of the differences in overall cell 'identity'. Therefore, similar analyses should be done for the total proteome and the transcriptome to determine whether the differences between samples are mostly restricted to RBPs or all widespread across a wide range of cellular pathways. If the second, which pathways are more affected.

Response: We concur with the reviewer's perspective that aggressiveness might not be distinctly discernible from the RNA interactome. We have therefore done PCA analysis based on new data of the total transcriptome and proteome of OS and GCTB cells, together with that of MSC and OB, and have included this analysis as Fig. 5c. As is evident from the PCA plots, the whole transcriptomes and proteomes of the more aggressive OSRH_2011/5 and I063_021 OS are closer, and distinct from that of the less aggressive OSKG and NRH_OS1, which also cluster closer to the MSC and OB samples. This indicates that the differences between the OS cells with differential aggressiveness and growth rates are

prevalent over the entire transcriptome and proteome. Interestingly, the PCA plots show the GCTB sample NRH_GCT1 to be distinct from the more aggressive OS cells at the level of gene expression, suggesting that the phenotypic similarity of high growth rate and aggressiveness may be contributed by different gene expression programs, which is supported by further experiments included in this revision. The analysis therefore indicates distinct trajectories of the more aggressive and less aggressive subsets of the OS and the GCTB diverging from a common cell origin. We have also performed new gene ontology (GO) analyses of the differentially expressed genes both at the transcriptome and full proteome levels and included the data in Fig 6c and 6d. Interestingly the GO analysis shows that terms related to RNA metabolism and translation are highly enriched in the transcriptomes and proteomes of aggressive OS, and to some extent in the GCTB, but are absent from the less aggressive OS. These data thus indicate a translation-centric gene expression program in the aggressive osteosarcomata, correlating with the enrichment of RBPs involved in mitochondrial and cytoplasmic translation in these OS (Fig 6a and 6b).

Functional outputs. While good OMICS data can be the basis for hypothesis-driven work, I feel some degree of functional validation is required to differentiate between causes and consequences. I understand such work cannot be applied to all the discoveries of a proteome-wide study, but can at least be applied to an example to illustrate the value of the dataset. Particularly, when there is some novelty in a given discovery.

Response: As recommended, we present new data aimed at elucidating the functional role of IGF2BP3, the most prominently enriched RNA-binding protein in osteosarcomas in comparison to normal osteoblasts and mesenchymal stemcells. As outlined in our response to reviewer #1, these functional analyses have unveiled a significant role for this RNA-binding protein in an oncogenic positive feedback loop with Myc (Fig. 8). Importantly, this loop demonstrates vulnerability to clinically approved drugs, thereby underscoring the translational potential of our novel discoveries.

Conclusion: the work is, in principle, technically sound and original in the sense of employing patient-derived cells. On the other hand, evidence of the role of RBP rewiring in inducing OS aggressiveness is missing. Functional validation would strengthen the conclusions and test the value of the dataset.

Response: We appreciate the reviewer's acknowledgment of the technical soundness and originality of our work, particularly in its utilization of patient-derived cells. We have endeavored to address the reviewer's concerns regarding the need for functional validation, as described above.

Minor points:

Page 5 lane 112 - the analysis has still 5% FDR, which means that 5 out of 100 proteins will be wrongly assigned. Therefore, bona fide sounds too adventurous and I would rather use 'high confidence' RBP. It would be different if the protein has been reported as RBP by different studies. In such case, there will be an accumulated high probability for the protein to be an RBP.

Page 6, lane 134 - has not been known > rephrase. 'remains unknown'?

Response: In accordance with the suggestion, we have revised the terminology used to describe the notably enriched proteins, shifting from "bona fide" to "high confidence" RBPs, and have rephrased the relevant terms on page 6 as recommended.

Figure 2b: Venn-diagram is missing.

Figure 3a: Bar plot is missing. Figure 3a: "The number of elevated and reduced RBPs are indicated in blue and orange respectively". The numbers in the figure have no color code.

Response: We appreciate the reviewer for bringing these formatting errors that occurred during the uploading process to our attention. We have now corrected these errors in Fig 2b and Fig 3a.

Reviewer #3 - Expertise in RBPs in cancer (Remarks to the Author):

This manuscript presents a characterization of the whole proteome and the RNA-binding proteome of osteosarcoma and bone tumor patient-derived cells, compared to normal mesenchymal stem cells and osteoblasts.

This characterization provides an interesting preliminary step in understanding the role of RNA-binding proteins in osteosarcoma, but further analyses and experiments would be essential to improve the significance of the results, link the observations to any pathogenic mechanism and identify promising therapeutic targets.

Specific points:

- In the introduction, the authors say: "In the current work, we comprehensively characterized RNA-RBP interactions in OS in an unbiased and systematic fashion by using enhanced RNA interactome capture (eRIC)". Yet the experiments are characterizing only the protein side, not the RNA side.

Parallel RNA-seq experiments would be crucial to understand whether the differential RNA-binding proteome is linked to alterations in RNA abundances and aberrant splicing of RNA isoforms (particularly relevant since the authors identify a group of splicing factors with altered RNA-binding). Although the technique will not capture individual RBP-RNA interactions, the characterization and comparison of altered protein binding sites on RNAs would be essential to identify pathogenic post-transcriptional RNA processes mediated by RNA binding proteins. Focusing on the protein side of the interaction is only half of the story.

Response: We appreciate this reviewer's concern and have now complemented our analysis of the RNA interactomes by an analysis of the transcriptomes along with a more detailed analysis of the total proteomes of the osteosarcoma and the normal bone and mesenchymal stem cells. Further, we have now integrated the differential mRNA expression data with the RBP interactome data by selecting specific RBPs, such as IGF2BP3, MEX3A and AKAP1, and determining the differential expression of their target RNAs (obtained from published literature or ENCODE datasets) in cell lines exhibiting differential enrichment of these RBPs. Remarkably, we see higher expression of mRNA targets of these RBPs in the transcriptomes of the cells that show greater enrichment of these RBPs in the eRIC data, suggesting a

correlation between enhanced RNA binding by these RBPs and enhanced expression of these target RNAs. We have also performed new gene ontology (GO) analyses of the differentially expressed genes both at the transcriptome and full proteome levels and included the data in Fig 6c and 6d. Interestingly the GO analysis shows that terms related to RNA metabolism and translation are highly enriched in the transcriptomes and proteomes of aggressive OS, and to some extent in the GCTB, but are absent from the less aggressive OS. These data thus indicate a translation-centric gene expression program in the aggressive osteosarcomata, correlating with the enrichment of RBPs involved in mitochondrial and cytoplasmic translation in these OS (Fig 6a and 6b).

However, we hope that this reviewer might agree with us that each RBP binds to numerous target RNAs, simultaneously with multiple RBPs binding to the same target RNAs, giving rise to complex post-transcriptional outcomes including splicing, RNA stability and translation, and that it is beyond the scope of this study to include a comprehensive transcriptome-wide characterization of the interacting targets of these RBPs and the functional consequences of such interactions.

- The identification of several RBPs involved in Stress Granule formation is potentially interesting, but it should be validated by additional evidence and dedicated experiments: can the authors see alterations in stress granule formation in osteosarcoma cells by microscopy approaches? Are osteosarcoma cells more resistant to stress? Are osteosarcoma cells more vulnerable to stress granule inhibitors? How is stress granule formation connected with the increased vulnerability to translation inhibition observed by the authors?

Response: It is indeed intriguing to note that several RBPs associated with stress granule assembly were detected in the eRIC analysis. As detailed above, our current functional investigations have been centered on elucidating an oncogenic positive feedback loop between IGFBP3 and Myc, which we have identified as susceptible to treatment with clinically approved drugs (as discussed both below and above in our response to reviewer #1). Consequently, we will undertake a future study specifically dedicated to exploring stress granule formation in the OS and shifted the section on RBPs involved in SGs to the Discussion section of the current manuscript.

- The authors show improved toxicity of cycloheximide in more aggressive osteosarcoma cells. As the authors say, this is only a proof of concept. Apart from exploring the mechanisms underlying alterations in cytoplasmic and mitochondrial translation, additional evidence with clinically feasible translation inhibitors (such as homoharringtonine, suggested by the authors) would be important to confirm the results and show that this vulnerability is therapeutically amenable.

Response: As suggested, we expanded our functional investigations on translation inhibition to homoharringtonine (HHT), a translation inhibitor that is FDA-approved for the treatment of chronic myeloid leukemia (CML). Confirming and extending our earlier data, the results demonstrate that HHT effectively impedes protein synthesis in the osteosarcoma (OS) cells (Fig. 7). Importantly, this inhibition occurs at therapeutic plasma concentrations and exhibits greater efficacy in the "translation-hungry" OS cell lines when compared to cell lines characterized by lower protein synthesis rates. It is noteworthy that this effect is specific to osteosarcoma cells with a high translation rate, as the highly aggressive and

translationally equally active malignant giant cell tumor (GCT) cells do not exhibit such a response to HHT.

- The work feels poorly connected to previous osteosarcoma literature. The authors in the abstract say: “Genomic, epigenomic and transcriptomic analyses have demonstrated the exquisite molecular complexity of this tumor, but haven’t sufficiently defined the underlying mechanisms or identified promising therapeutic targets”. How does this systematic analysis of RNA binding proteins relate to previous characterizations of osteosarcoma by omics approaches? A comparative analysis would improve the significance of the results by linking the observations presented in the manuscript to previous omics characterizations of osteosarcoma.

Response: We agree with this reviewer in that there are several omic studies of osteosarcoma. We have now highlighted in our introduction and discussion (p3 and p22) those studies that indicated differential gene expression pointing to oncogenic mechanisms in osteosarcoma. We have also stated more explicitly that the novelty of the current study lies in the analysis of the specific role of RNA-binding proteins in cells derived from primary tumor samples. Previously, there has only been a single published study analyzing RBPs in osteosarcoma using bioinformatic approaches to identify an RBP-related prognostic signature for OS. There is no previous experimental analysis connecting the RNA interactome with the transcriptome and proteome of osteosarcoma.

- Could differences in sample composition or cell heterogeneity act as a possible bias when comparing osteosarcoma samples with normal mesenchymal stem cells and osteoblasts, as well as when comparing different tumor samples?

Response: We concur with the reviewer's observation that many tumors exhibit intricate subclonal architectures. It remains possible that the tumor cells analyzed in this study originate from a particular subclone of the primary tumor. We now explicitly acknowledge this aspect in the Discussion section of our manuscript (p18).

- Is there any link between altered RNA-binding proteins and genomic lesions, copy number alterations and DNA methylation changes observed by the authors in osteosarcoma cells (Supplementary Figure 1)?

Response: We thank the reviewer for this intriguing question. In fact, in our revised manuscript, we have now experimentally established a connection between the genomic amplification of c-Myc, as depicted in Supplementary Fig. 1, and the RNA-binding protein IGF2BP3, which stands as the most prominently enriched RBP in the osteosarcoma (OS) cells we have analyzed. IGF2BP3 is a known post-transcriptional regulator of Myc expression, as it enhances both Myc mRNA stability and translation. Importantly, it has been demonstrated that Myc, in turn, functions as a transcriptional activator of IGF2BP3. These findings collectively suggest the presence of a positive feedback loop between IGF2BP3 and c-Myc, a hypothesis that we have experimentally validated in the work presented here.

Specifically, by inhibiting IGF2BP3 expression, we observed a consequent downregulation of c-Myc expression, and conversely, when c-Myc was pharmacologically inhibited, it led to downregulation of IGF2BP3 expression. This observation underscores the existence of an additional layer of post-

transcriptional regulation mediated by RNA-binding proteins, which exerts influence over altered gene expression resulting from genomic lesions and gene amplifications.

- PCA/MDS plots of samples based on total proteomics and RNA-binding proteomics measurements would be important to capture and visualize overall similarities/differences among total protein and RNA interactome samples between tumor cells, normal mesenchymal stem cells and osteoblasts.

Response: As suggested, we have now generated PCA plots based on total transcriptomes, proteomes and RNA interactomes OS and GCTB cells, together with those of MSC and OB (Fig. 5c). These PCA plots show that the whole transcriptomes and proteomes of the more aggressive OSRH_2011/5 and I063_021 OS are closer, and distinct from that of the less aggressive OSKG and NRH_OS1, which also cluster closer to the MSC and OB samples. Interestingly, the PCA plots show the GCTB sample NRH_GCT1 to be distinct from the more aggressive OS cells at the level of gene expression, suggesting that the phenotypic similarity of high growth rate and aggressiveness is likely contributed by different gene expression programs, which is supported by further experiments that we have performed for this revision.

- Some of the figures (e.g. 2B or 3A) had partially missing graphical elements in the manuscript pdf file. For example, in Figure 3A, I could only see numbers, but not bars. I would check for possible compatibility issues due to different operating system software.

Response: We thank both this reviewer and reviewer #2 for bringing this error to our attention. We have now taken care to check the integrity of the figures following the uploading process.

I thank the authors for the hard work on this.

Response: We value the reviewer's acknowledgment of the substantial work by several scientists over a number of years to undertake this study.

Reviewers' Comments:

Reviewer #1:

Remarks to the Author:

Thank you for adding the translation and slightly more mechanistic efforts with IGF2BP3. I still would consider these to be overall novel results that are of interest, albeit from a necessarily small sample size.

Reviewer #2:

Remarks to the Author:

Authors have answered most of my questions satisfactorily.

Only a main point remains: I think it is important to see the fold changes in scatter plots and fold change and p-values between eRIC/eRIC, eRIC/proteome, eRIC/transcriptome and proteome/transcriptome for each cell group and comparison. This has two purposes, quality control and analysis of the relationship between "classification as enriched in one group" and binding tendency (eRIC vs eRIC), global RNA binding avidity (eRIC vs proteome) and relationships between substrates and RBPs (eRIC/transcriptome). As indicated above, cut offs are ok to generate high confidence hits, however, if it is not good for clustering and dynamics profiling. E.g. A protein displays a fold change of 0.54 in one line and 0.62 in other. The binding propensity is very similar, but in one case passed the threshold. Therefore, one must take this case with caution. Another example is a protein with a fold change of 0.54 in one comparison and 0.8 in other; if they reflect different levels of aggressiveness these results might indicate a trend. However, by ignoring the fold changes under 0.6 this could not be observed. Moreover, the ratio eRIC/total proteome is a reflection of crosslinkability, which at the same time is dependent on the time a protein sits on the RNA. Therefore, it would be possible to calculate the "binding activity" by normalizing for protein abundance (allowing to distinguish between abundance-driven changes and activity-dependent changes). This has been done before. Finally, the RNA seq can be used to explore changes in substrate availability by exploiting the eCLIP datasets. The latest would be nice, although I don't see it as critical. I think the datasets provided offer a great opportunity to identify RBPs with activity and abundance driven changes.

Reviewer #3:

Remarks to the Author:

The authors addressed all the major issues of their original submission, complementing their results with transcriptome analyses, expanding functional investigations with translation inhibitors and characterizing the MYC-IGF2BP3 feedback loop.

Remaining considerations:

One overall consideration is the specificity to osteosarcoma of the MYC-IGF2BP3 axis reported by the authors, considering that IGF2BP3 and MYC are upregulated and amplified, respectively, in many tumor types. This specificity, or absence thereof, should be addressed in the discussion.

In the introduction, the authors state: "recent bioinformatics-based meta-analysis, utilizing gene expression data from OS samples and clinical data from osteosarcoma patients, generated an RBP-related prognostic signature for OS consisting of seven hub RBPs". The authors should link their results to this previous signature in the discussion (how many of the 7 hub RBPs overlap with the ones they identify?).

In the results, the authors say: "A total number of 6134 and 593 proteins were identified in the FPs and eRIC eluates, respectively (Supplementary Data 1). Proteins that were significantly enriched (at least 2-fold enrichment, $\log_2 FC \geq 1$) and an FDR < 0.05) in UV-crosslinked compared to no-UV controls were considered high confidence RBPs (Fig. 2a, Supplementary Data 1). We identified a total of 593 RBPs from all cells studied (OS, OB, MSCs), which we refer to as the bone/mesenchymal-cell RNA interactome". Does this mean that all the 593 proteins identified in

eRIC eluates had $\log_2 FC \geq 2$ and an $FDR < 0.05$, and none was filtered out according to fold change or FDR thresholds? The fact that all detected proteins are significantly enriched raises concerns about the analysis steps (possibly a consequence of calculating different normalization coefficients for UV and noUV samples?)

In Figure 5C, PCA plots based on the total proteome or eRIC are similar. Is this something expected?

In some figures, such as Supplementary Figures 3 and 4, the text is too small to be readable.

Transcriptome processed data, such as FPKM values for each gene in each sample, should be available in a supplementary data file (similar to Supplementary Data File 1, containing all processed proteomics and eRIC data).

It would be important to understand why certain cells with high translation rates are responsive to HTT (OS cells) and others are not (GCTB cells, although GCTB is represented only by a single sample).

Reviewer #2:

Authors have answered most of my questions satisfactorily

1. *Only a main point remains: I think is important to see the fold changes in scatter plots and fold change and p-values between eRIC/eRIC, eRIC/proteome, eRIC/transcriptome and proteome/transcriptome for each cell group and comparison. This has two purposes, quality control and analysis of the relationship between “classification as enriched in one group” and binding tendency (eRIC vs eRIC), global RNA binding avidity (eRIC vs proteome) and relationships between substrates and RBPs (eRIC/transcriptome).*

As recommended by the reviewer, we have generated scatterplots for eRIC vs transcriptome along with Volcano plots for eRIC, proteomic and transcriptomic data compared to osteoblast controls, now depicted in Supplementary Figures 3, 4, 5 and 7. Regarding the additional suggestion to illustrate fold changes and p-values for comparisons between eRIC/eRIC, eRIC/proteome, eRIC/transcriptome, and proteome/transcriptome across each cell group, we think that the benefits would not outweigh the disadvantages, because it would necessitate the creation of 168 scatterplots (5 OS cell groups + 2 control cells (OB & MSC), each compared against the other (7x7) x 4 comparisons – 7x4 self-comparisons (196-28=168)). We believe that presenting our data in this manner would not be conducive to reader comprehension, as it could be challenging to maintain focus and convey meaningful interpretations.

1. *As indicated above, cut offs are ok to generate high confidence hits, however, it is not good for clustering and dynamics profiling. E.g. A protein displays a fold change of 0.54 in one line and 0.62 in other. The binding propensity is very similar, but in one case passed the threshold. Therefore, one must take this case with caution. Another example is a protein with a fold change of 0.54 in one comparison and 0.8 in other; if they reflect different levels of aggressiveness these results might indicate a trend. However, by ignoring the fold changes under 0.6 this could not be observed.*

We concur with the reviewer's perspective that even subtle variations in protein enrichment between tumor and normal cells and between the different tumor cells may possess functional significance. Simultaneously, we acknowledge the necessity for a meticulous interpretation of the data, emphasizing significant changes. In the scatter-plots comparing eRIC and the full proteome (FP), we employ a combination of log₂ fold change and p-values (color coding) to depict proteins exhibiting noteworthy alterations. However, recognizing the potential

importance of the complete datasets, devoid of imposed cutoffs, to the community of researchers, we have included the complete datasets in supplementary data files (Supplementary data 8)

3. Moreover, the ratio eRIC/total proteome is a reflection of crosslinkability, which at the same time is dependent on the time a protein sits on the RNA. Therefore, it would be possible to calculate the "binding activity" by normalizing for protein abundance (allowing to distinguish between abundance-driven changes and activity-dependent changes). This has been done before.

We concur with the reviewer's observation that the ratio of eRIC to proteome may offer valuable insights into the "binding activity" of a given RNA-binding protein (RBP). However, due to technical constraints, eRIC and full proteome samples had to be assayed separately in distinct Tandem Mass Tag (TMT) sets. As the batch effects surpass the variability within a TMT set, normalizing with values from different measurements/TMT sets is likely to be unreliable. Nevertheless, in accordance with the reviewer's recommendation, we have introduced an additional column in Supplementary Tables 7, delineating the ratio of enrichment in eRIC to abundance in the full proteome. This ratio can be regarded as a "measured RNA binding activity" for the respective RBPs. Furthermore, we have incorporated this estimation of binding activity into the results section (p10).

4. Finally, the RNA seq can be used to explore changes in substrate availability by exploiting the eCLIP datasets. The latest would be nice, although I don't see it as critical. I think the datasets provided offer a great opportunity to identify RBPs with activity and abundance driven changes.

We agree that the transcriptomic data can be integrated with the eCLIP datasets of various RBPs which are available in databases such as ENCORE to explore changes in RNA availability. However, this is a task which we consider to be beyond the scope of this manuscript especially because any of these data would require experimental validation for each RBP eCLIP dataset. The interpretation of such data would be further confounded, because eCLIP datasets are derived from standard cell lines whereas our eRIC and transcriptomic datasets are from patient-derived tumour cells.

REVIEWER #3

The authors addressed all the major issues of their original submission, complementing their results with transcriptome analyses, expanding functional

investigations with translation inhibitors and characterizing the MYC-IGF2BP3 feedback loop.

Remaining considerations:

One overall consideration is the specificity to osteosarcoma of the MYC-IGF2BP3 axis reported by the authors, considering that IGF2BP3 and MYC are upregulated and amplified, respectively, in many tumor types. This specificity, or absence thereof, should be addressed in the discussion.

We agree with the reviewer that upregulation of IGF2BP3 and amplification of the Myc gene has previously been shown in other tumor types. A positive feedback loop involving these two proteins has also been attested in a study in neuroblastoma. However, our results implicate this positive feedback loop between the RNA-binding protein IGF2BP3 and the multifunctional transcription factor c-Myc for the first time in osteosarcoma thus expanding the implications in cancer biology and therapy. As recommended by the reviewer, we have now addressed this point more specifically in the discussion (p 23).

In the introduction, the authors state: "recent bioinformatics-based meta-analysis, utilizing gene expression data from OS samples and clinical data from osteosarcoma patients, generated an RBP-related prognostic signature for OS consisting of seven hub RBPs". The authors should link their results to this previous signature in the discussion (how many of the 7 hub RBPs overlap with the ones they identify?).

The RBPs found to be significantly enriched or reduced in the RNA interactomes of the OS in our analysis do not exactly overlap with 7 hub RBPs identified as a prognostic signature in the bioinformatic analysis. However, the RBPs identified in our analysis included IGF2BP3 and IGF2BP1 which are closely related to IGF2BP2 which is identified as one of the 7 hub RBPs. Similarly, our analysis has identified a zinc finger domain protein ZCCHC3, similar to ZC3HAV1 identified as one of the 7 hub RBPs in the bioinformatic analysis. Also, one of the 7 hub RBPs identified was RBM34 whereas a number of RBM proteins were found to be either significantly elevated or reduced in the OS RNA interactomes in our analysis. This suggests that similar groups of RBPs, such as IGF2BPs, zinc finger proteins and RBMs, might be dysregulated across OS RNA interactomes, with specific RBPs among these types showing specific alterations in the interactomes of different osteosarcomata. We have now discussed these points more explicitly in the discussion (p19).

In the results, the authors say: "A total number of 6134 and 593 proteins were identified in the FPs and eRIC eluates, respectively (Supplementary Data 1). Proteins that were significantly enriched (at least 2-fold enrichment, $\log_2 FC \geq 1$) and an FDR <0.05) in UV-crosslinked compared to no-UV controls were considered high confidence RBPs (Fig. 2a, Supplementary Data 1). We identified a total of 593 RBPs from all cells studied (OS, OB, MSCs), which we refer to as the bone/mesenchymal-cell RNA interactome". Does this mean that all the 593 proteins identified in

eRICeluates had $\log_2 FC \geq 2$ and an $FDR < 0.05$, and none was filtered out according to fold change or FDR thresholds? The fact that all detected proteins are significantly enriched raises concerns about the analysis steps (possibly a consequence of calculating different normalization coefficients for UV and noUV samples?)

The total of 593 RBPs which were identified as the bone/mesenchymal cell RNA interactome is the sum of the RBPs which were found to be significantly enriched ($\log_2 FC \geq 1$ and an $FDR < 0.05$) in the OS, OB and MSC after filtering, represented by the union of the sets shown in Fig. 2b. These were obtained after filtering the proteins obtained in the eRIC eluates which did not satisfy these criteria, which are shown at the bottom of the columns in the tables in Supplementary data 1 (Column Q (hit) = False).

In Figure 5C, PCA plots based on the total proteome or eRIC are similar. Is this something expected?

The PCA plots based on the transcriptome, proteome and eRIC in Figure 5C are broadly similar, demonstrating fundamental differences in the gene expression programs between the more aggressive and less aggressive OS cell types and the GCTB cells, which are reflected in all the three PCA plots. The PCA plots also indicate distinct evolutionary trajectories of divergence from the normal cells of origin in the more aggressive and less aggressive subtypes of the OS and the GCTB. This is expected as overall the enrichment of RBPs in the RNA interactomes correlates with their abundance in the proteomes (as also seen in Figures 6a and b) with notable exceptions which are specifically pointed out in the non-correlated groups in Figure 4.

In some figures, such as Supplementary Figures 3 and 4, the text is too small to be readable.

This has been changed by including larger versions of the heatmaps on different pages to improve the legibility of the gene names.

Transcriptome processed data, such as FPKM values for each gene in each sample, should be available in a supplementary data file (similar to Supplementary Data File 1, containing all processed proteomics and eRIC data).

This has been done by including the processed data from the transcriptome analysis as Supplementary data 9.

It would be important to understand why certain cells with high translation rates are responsive to HTT (OS cells) and others are not (GCTB cells, although GCTB is represented only by a single sample).

We agree that it is indeed of interest that the GCTB cells, which have high translation activity similar to the aggressive OS cells, do not exhibit the same vulnerability to the translation inhibitors. This firstly indicates that the response to HHT does not represent a mere non-specific effect on translationally active tumor cells, which we have already mentioned in p15 of the manuscript. Additionally, it is to be noted that the expression of Myc is low and of IGF2BP3 is undetectable in the GCTB cells (Fig. 8c), unlike that in the two aggressive OS cell types. This may indicate that the absence of the IGF2BP3/c-Myc positive feedback loop in the GCTB cells, and other possible differences in post-transcriptional regulatory programs, may contribute to the differential response of this aggressive bone tumor to translation inhibitors. We have now discussed this in p23-24 of the manuscript. Future studies may be directed at deciphering how these potential differences in post-transcriptional regulation may contribute to differential responses to therapy.